# Accelerated and Stable Convergence with Anchored Optimistic Method

**Motahareh Sohrabi** [1 2]  **Jianxin You** [1]  **Simon Lacoste-Julien** [1 2 3]  **Eduard Gorbunov** [4]  **Gauthier Gidel** [1 2 3]

## Abstract

We study first-order methods for solving monotone variational inequalities arising in min-max optimization. Classical approaches such as the extragradient method rely on two gradient queries per iteration, which limits their analysis and applicability in the online and stochastic settings. We propose a family of Generalized Optimistic Methods with Anchoring (GOMA), which combine two-time-scale optimistic updates with an anchoring term inspired by Halpern iteration. In the deterministic setting, GOMA achieves the optimal accelerated last-iterate rate $\mathcal{O}(1/k^2)$ on the squared gradient norm for monotone Lipschitz operators. In the stochastic setting with unbounded variance, a simplified single-call variant of GOMA achieves a last-iterate convergence rate of $\mathcal{O}(1/\sqrt{k})$ on the squared gradient norm. To the best of our knowledge, this is the first such guarantee for stochastic monotone Lipschitz VIs in the unconstrained setting without variance reduction or growing batches.

## 1. Introduction

Minimax optimization and more generally, Variational Inequality (VI) problems, naturally arise in adversarial training, constrained optimization and multi-agent reinforcement learning, where the goal is to find equilibrium solutions under structured interaction of agents or competing objectives. When solving VIs classical gradient descent fails to converge even in simple bilinear games (Mertikopoulos et al., 2019). A breakthrough came with the extragradient method (EG) of Korpelevich (1976), which introduces a correction step and guarantees convergence under monotonicity. However, (i) EG requires two gradient evaluations per iteration, which is computationally expensive and makes it

impractical in online or stochastic environments (Golowich et al., 2020). Moreover, subsequent work revealed that (ii) EG may fail in adversarial or stochastic regimes, motivating two-time-scale methods, such as DSEG algorithm (Hsieh et al., 2020). Above all (iii) EG has a last-iterate convergence guarantee of $O(1/k)$ for Monotone and Lipschitz operator in terms of operator norm, which is not optimal.

The optimistic method (Popov, 1980) reduces the per-iteration cost to a single gradient call by leveraging past gradients, addressing (i). Generalized optimistic method is a variant of optimistic method with two time-scales, addressing (ii). Also, anchoring, drawing inspiration from the Halpern fixed-point iteration (Halpern, 1967; Lieder, 2020), has recently emerged as a powerful mechanism for accelerating VI algorithms, addressing (iii). In this paper we introduce Generalized Optimistic Method with Anchoring (GOMA) which addresses these three issues simultaneously.

Our contributions are:

- We introduce GOMA, combining two-time-scale optimistic updates with Halpern-type anchoring.

- In the deterministic setting with monotone Lipschitz operators, GOMA attains an accelerated last-iterate convergence rate of $\mathcal{O}(1/k^2)$ in the squared operator norm, matching the complexity lower bound.

- In the stochastic setting, a simplified single-call variant of GOMA achieves a last-iterate convergence rate of $\mathcal{O}(1/\sqrt{k})$ on the squared operator norm under state-dependent noise. To the best of our knowledge, this is the first such guarantee for stochastic monotone Lipschitz VIs in the unconstrained setting without variance reduction or growing batches.

## 2. Preliminaries

Given a vector field $G : \mathbb{R}^d \to \mathbb{R}^d$, we study unconstrained *variational inequality* (VI) problems defined as:

$$\text{find } x^\star \in \mathbb{R}^d \quad \text{such that} \quad G(x^\star) = 0. \qquad \text{(VI)}$$

Throughout the paper, we measure convergence using the *last-iterate residual* $\|G(x_k)\|^2$.

---

[1]Université de Montréal [2]Mila - Quebec AI Institute [3]CIFAR AI Chair [4]Mohammed Bin Zayed University of Artificial Intelligence. Correspondence to: Motahareh Sohrabi <motahareh.sohrabi@mila.quebec>.

*Proceedings of the $43^{rd}$ International Conference on Machine Learning*, Seoul, South Korea. PMLR 306, 2026. Copyright 2026 by the author(s).

**Assumptions.** $G$ is *monotone* and *L-Lipschitz*:

$$\langle G(x) - G(y),\ x - y \rangle \geq 0, \qquad \forall x, y \in \mathbb{R}^d,$$
$$\|G(x) - G(y)\| \leq L\|x - y\|, \quad \forall x, y \in \mathbb{R}^d.$$

These assumptions characterize the standard class of monotone variational inequalities studied in first-order methods (Korpelevich, 1976; Nemirovski, 2004).

**Saddle-point problems.** A central example of (VI) arises from saddle-point (min–max) optimization. We consider

$$\min_{x \in \mathbb{R}^d} \max_{y \in \mathbb{R}^d}\ f(x, y),$$

where $f : \mathbb{R}^d \times \mathbb{R}^d \to \mathbb{R}$ is continuously differentiable. Define $z = (x, y) \in \mathbb{R}^d \times \mathbb{R}^d$, and introduce the *gradient operator*

$$G(z) := \begin{pmatrix} \nabla_x f(x, y) \\ -\nabla_y f(x, y) \end{pmatrix}.$$

Under the monotonicity of $G$ (equivalently, $f$ convex–concave), $z^\star = (x^\star, y^\star)$ is a saddle point if and only if $G(z^\star) = 0$. So solving the saddle-point problem is equivalent to solving the variational inequality problem (VI).

In modern machine-learning applications of saddle-point problems, the variables $z = (x, y)$ parameterize neural networks and the saddle point lies in unbounded Euclidean space, making the squared operator norm $\|G(z_k)\|^2$ the de facto stationarity measure. By contrast, classical algorithmic-game-theory settings with intrinsic bounded strategy spaces (e.g., matrix games on the simplex) use the gap function $\mathrm{GAP}(z) = \sup_{z' \in X} \langle G(z), z' - z \rangle$ (Nesterov, 2007) as the progress measure. The gap function requires a bounded domain, since the supremum diverges on $\mathbb{R}^d$, and gives rise to fundamentally different analyses (Cai et al., 2022b; Abe et al., 2025) and is suitable for constrained games' convergence analysis.

## 3. Related Work

### 3.1. Algorithms for Solving Variational Inequalities

Solving variational inequalities (VI), with standard gradient descent can exhibit oscillatory behavior (Platt & Barr, 1987; Gidel et al., 2019b). A classical algorithm for addressing this behavior is the extragradient method (Korpelevich, 1976), given by

$$\begin{aligned} y_k &= x_k - \eta_k G(x_k) \\ x_{k+1} &= x_k - \eta_k G(y_k). \end{aligned} \qquad \text{(EG)}$$

For monotone Lipschitz operators, (EG) achieves an *ergodic* (average) rate of $O\left(\frac{1}{k}\right)$ duality gap (Nesterov, 2007). This rate is optimal and matches the lower bound of $O\left(\frac{1}{k}\right)$ from Nemirovski (2004).

However, a method may have an *ergodic* convergence rate but no *last-iterate* convergence: finite regret ensures convergence of the ergodic averages, while the iterates themselves may cycle indefinitely and not converge (Bailey et al., 2020).

The last-iterate convergence rate of extragradient in terms of squared operator norm for the same class of operators is $O\left(\frac{1}{k}\right)$ (Gorbunov et al., 2022a). This rate is not optimal, as the lower bound for this class is $O\left(\frac{1}{k^2}\right)$ (Yoon & Ryu, 2021). As a result there exist several accelerated methods that achieve a rate of $O\left(\frac{1}{k^2}\right)$ (Yoon et al., 2024; Lee & Kim, 2021; Tran-Dinh & Luo, 2021).

Despite their favorable convergence properties, extragradient methods rely on two operator evaluations per iteration. This is misaligned with the online-learning setting, which provides only a single gradient at the chosen action (Golowich et al., 2020). Optimistic gradient methods address this limitation by using extrapolation from past gradients rather than additional oracle queries (Golowich et al., 2020; Popov, 1980).

$$\begin{aligned} y_k &= x_k - \eta_k G(y_{k-1}) \\ x_{k+1} &= x_k - \eta_k G(y_k). \end{aligned} \qquad \text{(OM)}$$

The optimistic gradient method achieves an $O\left(\frac{1}{k}\right)$ last-iterate convergence rate for monotone Lipschitz operators (Gorbunov et al., 2022b; Cai et al., 2022b). Several works have proposed ways to accelerate this rate, including Tran-Dinh & Luo (2021). Our work contributes to this line of research by studying *last-iterate* acceleration for a class of *optimistic gradient based* methods and providing stochastic convergence analysis.

### 3.2. Two-Time-Scale Methods

To improve last-iterate convergence and stability, two-time-scale strategies were introduced for extragradient-type dynamics in stochastic regimes where single-scale methods may fail to converge. In particular, Hsieh et al. (2020) showed that, in the stochastic setting, using a larger step size for the extrapolation step than for the correction step prevents the failure of convergence of extragradient and yields almost sure last-iterate convergence at a rate of up to $O\left(\frac{1}{k}\right)$ in affine problems.

Subsequently, Lee & Kim (2021) proposed the Fast Extragradient (FEG) method, which extends the two-time-scale idea to smooth problems under a negative comonotonicity assumption, achieving an accelerated $O\left(\frac{1}{k^2}\right)$ rate and stochastic convergence guarantees with growing batch-size. Together, these results show that time-scale separation is an effective mechanism for stabilizing and accelerating extragradient methods.

The same principle can be applied to optimistic, single-query methods. Accordingly, Mokhtari et al. (2020) introduced

the generalized optimistic method, which allows separate step sizes for the prediction and correction steps,

$$y_k = x_k - \gamma_k G(y_{k-1}),$$
$$x_{k+1} = x_k - \eta_k G(y_k). \quad \text{(Generalized OM)}$$

In parallel, Stooke et al. (2020) and Sohrabi et al. (2024) showed the effectiveness of proportional–integral (PI) controllers for solving the Lagrangian saddle-point formulation of constrained optimization problems. Sohrabi et al. (2024) further showed that this PI-controller dynamics is equivalent to the generalized optimistic method of Mokhtari et al. (2020), revealing two-time-scale optimistic algorithms as an effective feedback-control system.

### 3.3. Halpern-Type Acceleration

A principal mechanism for *accelerating* first-order methods in monotone variational inequalities is Halpern-type anchoring. The Halpern method (Halpern, 1967) was originally proposed for solving fixed-point problems

$$z = T(z),$$

where $T : \mathbb{R}^d \to \mathbb{R}^d$ is a nonexpansive operator. Its classical iteration is

$$z_{k+1} = \beta_k z_0 + (1 - \beta_k) T(z_k),$$

where $\beta_k \in (0, 1)$ decreases to zero. To solve monotone variational inequalities $G(z) = 0$, with $G(z)$ is the gradient operator, a standard construction is to take $T = (I + \alpha G)^{-1}$, the resolvent of $G$, which is firmly nonexpansive when $G$ is monotone (Bauschke & Combettes, 2020). However, computing the resolvent is generally significantly more expensive than an explicit update using $G$ and therefore is outside the scope of this paper, which is focused on first-order methods.

Modern algorithms therefore use a *Halpern-type anchoring* written directly in terms of the operator $G$, leading to the explicit update (Anchoring). This formulation can be viewed as a first-order realization of the classical Halpern iteration, as explained in (Diakonikolas, 2020) which is the anchoring mechanism used throughout this paper.

$$z_{k+1} = z_k - \alpha_k G(z_k) + \beta_k(z_0 - z_k). \quad \text{(Anchoring)}$$

**Why anchoring helps.** The anchoring update can be interpreted as a gradient step on a *regularized* operator. Define

$$\widetilde{G}_k(z) = G(z) + \frac{\beta_k}{\alpha_k}(z - z_0).$$

Then (Anchoring) can be written as a forward step with respect to $\widetilde{G}_k$:

$$z_{k+1} = z_k - \alpha_k \widetilde{G}_k(z_k),$$

where, $\widetilde{G}_k$ is $\frac{\beta_k}{\alpha_k}$-strongly monotone:

$$\langle \widetilde{G}_k(x) - \widetilde{G}_k(y),\ x - y \rangle$$
$$= \langle G(x) - G(y),\ x - y \rangle + \frac{\beta_k}{\alpha_k}\|x - y\|^2\ \geq\ \frac{\beta_k}{\alpha_k}\|x - y\|^2.$$

Thus, the anchoring term endows the operator with an *artificial strong monotonicity* that accelerates convergence. As $\beta_k \downarrow 0$, this regularization vanishes, recovering the original monotone problem while providing acceleration in the transient regime.

**Connection to weight decay.** The regularizer $\frac{\beta_k}{\alpha_k}(z - z_0)$ in $\widetilde{G}_k$ is an $\ell_2$ (weight-decay) penalty (Krogh & Hertz, 1991; Loshchilov & Hutter, 2019), but centered at the initialization $z_0$ rather than the origin and applied with a vanishing coefficient. A constant weight-decay coefficient would shift the fixed point and bias the solution toward $z_0$; the decay $\beta_k \downarrow 0$ instead removes this bias asymptotically. Anchoring therefore provides the transient stabilization of weight decay, the artificial strong monotonicity noted above, while still converging to a solution $G(z^\star) = 0$ of the original problem.

Anchoring mechanism has emerged as the key mechanism for last-iterate acceleration of variational inequalities. Methods like *Extra Anchored Gradient (EAG)* algorithm (Yoon & Ryu, 2021), *Fast Extragradient (FEG)* algorithm (Lee & Kim, 2021), and *Anchored Popov* (Tran-Dinh & Luo, 2021) all use anchoring mechanism to achieve acceleration.

Anchoring has also been applied in reinforcement learning, where it provides stability and accelerates convergence. Sokota et al. (2023) shows that anchoring connects RL, quantal response equilibria, and zero-sum games by damping oscillations and guiding updates toward equilibria. More recently, anchoring has been used to accelerate value iteration (Lee & Ryu, 2023), yielding faster convergence without sacrificing optimality. These results highlight anchoring as a general mechanism for stabilizing and accelerating learning in sequential decision-making.

## 4. Generalized Optimistic Method with Anchoring (GOMA)

In this section, we introduce the *Generalized Optimistic Method with Anchoring (GOMA)*, a family of algorithms that equips classical optimistic gradient method (OM) with two-time-scale update (Generalized OM) and the anchoring mechanism (Anchoring).

**Generalized Optimistic Method with Anchoring (GOMA)**

$$y_k = \beta_k x_0 + (1 - \beta_k)x_k - \gamma_k G(y_{k-1}),$$
$$x_{k+1} = \beta_k x_0 + (1 - \beta_k)x_k - \eta_k G(y_k). \quad \text{(GOMA)}$$

Here, $\gamma_k$ and $\eta_k$ denote the step-sizes for the exploration

and update steps, respectively. The coefficient $\beta_k \in [0, 1)$ is the anchoring parameter, which gradually decays to zero as $k \to \infty$. Throughout this paper, we assume

$$\beta_k = \frac{a}{k+b} \quad \text{for} \quad b > a.$$

Several well-known algorithms arise as special cases of GOMA:

- Setting $\beta_k = 0$ recovers the *generalized optimistic method* (Mokhtari et al., 2020).

- Setting $\gamma_k = \eta_k$ yields the *anchored Popov algorithm* (Tran-Dinh & Luo, 2021).

- Setting both $\beta_k = 0$ and $\gamma_k = \eta_k$ reduces the scheme to *classical Popov method* (Popov, 1980), also known as the *optimistic method* or *past extragradient (PEG)*.

### 4.1. Proof Outline

To prove the convergence of (GOMA), we follow the standard Lyapunov analysis and construct a potential function, which will serve as the basis for the descent argument of $\|G(x_k)\|$. Define the Lyapunov function:

$$V_k = a_k \|G(x_k)\|^2 + b_k \langle G(x_k), x_k - x_0 \rangle \\ + c_k L^2 \|x_k - y_{k-1}\|^2, \quad (1)$$

where $a_k = c_k = \frac{b_k \eta_k}{2 \beta_k}$, and $b_{k+1} = \frac{b_k}{1 - \beta_k}$.

To simplify hyperparameter tuning while preserving the benefits of a two-time-scale design, we consider two parameter setups: (I) we fix the exploration step size to $\gamma_*$ and set the update step size to $\eta_k = (1 - \beta_k)\gamma_*$; and (II) we fix the exploration step size to $\eta_*$ and set the update step size to $\gamma_k = (1 - \beta_k)\eta_*$. Each setup may be preferable depending on whether a larger exploration step size or a larger update step size is desired.

#### Case I: larger update step.

We first study the schedule with a larger update step-size, $\eta_k = \eta_*$ and exploration scaled by $(1 - \beta_k)$. The next lemma states that the one-step Lyapunov decrease holds whenever $(\eta_k, \beta_k)$ satisfy three elementary conditions that arise from bounding Lipschitz and monotonicity cross-terms.

**Lemma 1** (One-step Lyapunov decrease)**. [Proof in Appx. A.1.] *Let $G$ be monotone and $L$-Lipschitz, and consider the iterates*

$$y_k = \beta_k x_0 + (1 - \beta_k)x_k - \eta_*(1 - \beta_k)G(y_{k-1}), \\ x_{k+1} = \beta_k x_0 + (1 - \beta_k)x_k - \eta_* G(y_k). \quad (\triangle)$$

*We prove that the Lyapunov function (1) is decreasing if the step-size $\eta_k$ satisfies the following conditions:*

$$\eta_{k+1} \leq \frac{\beta_{k+1}}{2M \eta_k \beta_k (1 - \beta_k)}, \quad (2)$$

$$1 - 2M\eta_k^2(1 - \beta_k)^2 - M\eta_k^2 \beta_k^2 \geq 0, \quad (3)$$

$$\eta_{k+1} \leq \frac{\beta_{k+1}}{2\beta_k(1-\beta_k)}\Big[\frac{2(1-\beta_k^2)-4M\eta_k^2(1-\beta_k)^2-\beta_k^2}{1-2M\eta_k^2(1-\beta_k)^2-M\eta_k^2\beta_k^2}\eta_k\Big], \quad (4)$$

*for $M := 2L^2(1 + \theta)$ and $\theta \geq 0$. With $\widetilde{c}_{k+1} \geq 0$ for $\theta = 2$, these give:*

$$V_k - V_{k+1} \geq \widetilde{c}_{k+1} L^2 \|x_{k+1} - y_k\|^2.$$

*Conditions of Eq. (2), Eq. (3), Eq. (4) are satisfied, with constant step-size $\eta_k = \eta_* \in (0, \frac{1}{2\sqrt{3}L})$, and the choice of $\beta_k = \frac{2}{k+6}$.*

**Theorem 1.** [Proof in Appx. A.2.] *Suppose $G$ is monotone and $L$-Lipschitz continuous. Consider the updates of Eq. ($\triangle$) With the parameter choices*

$$\beta_k = \frac{2}{k+6}, \quad \eta_* \in \Big(0, \tfrac{1}{2\sqrt{3}L}\Big),$$

$$a_k = c_k = \frac{b_0 \eta_*}{80}(k+4)(k+5)(k+6),$$

$$b_k = \frac{b_0}{20}(k+4)(k+5)$$

*the Lyapunov decrease from Lemma 1 implies the bound*

$$\|G(x_k)\|^2 \leq \frac{16/\eta_*^2 + 72L^2}{(k+6)^2} \|x_0 - x^\star\|^2. \quad (5)$$

*Moreover, the constant $16/\eta_*^2 + 72L^2$ is decreasing in $\eta_*$, so it is smallest at the largest step; as $\eta_* \to \frac{1}{2\sqrt{3}L}$ it gives*

$$\|G(x_k)\|^2 \leq \frac{264L^2}{(k+6)^2} \|x_0 - x^\star\|^2. \quad (6)$$

The bound (5) gives an $O(1/k^2)$ decay of the residual $\|G(x_k)\|^2$ with explicit constants and a single scalar hyperparameter $\eta_*$.

**Case II: larger exploration step.** We next analyze the complementary schedule with a larger exploration step-size, keeping the update scaled by $(1 - \beta_k)$. This variant can be preferable when exploration requires a larger look-ahead while updates must remain conservative.

**Lemma 2.** [Proof in Appx. A.3.] *Let $G$ be monotone and $L$-Lipschitz, and consider the iterates*

$$y_k = \beta_k x_0 + (1 - \beta_k)x_k - \gamma_* G(y_{k-1}), \\ x_{k+1} = \beta_k x_0 + (1 - \beta_k)x_k - (1 - \beta_k)\gamma_* G(y_k). \quad (\square)$$

*The potential in (1) is non-increasing for the algorithm if the following conditions are satisfied.*

$$\gamma_{k+1} \leq \frac{\beta_{k+1}}{2M \beta_k \gamma_k} \frac{(1 - \beta_k)^2}{(1 - \beta_{k+1})}, \quad (7)$$

$$1 - 2M\gamma_k^2 - M\beta_k^2\gamma_k^2 \geq 0, \qquad (8)$$

$$\gamma_{k+1} \leq \frac{\beta_{k+1}}{2\,\beta_k(1-\beta_{k+1})}\left[\frac{4M\gamma_k^2 + \beta_k^4 - 2\beta_k^3 + 3\beta_k^2 - 2}{(1-\beta_k)^2\left(M(\beta_k^2+2)\gamma_k^2-1\right)}\right] \qquad (9)$$

*for $M := 2L^2(1+\theta)$ and $\theta \geq 0$. With $\widetilde{c}_{k+1} \geq 0$ for $\theta = 2$, these give:*

$$V_k - V_{k+1} \geq \widetilde{c}_{k+1}L^2\|x_{k+1}-y_k\|^2.$$

*Conditions of Eq. (7), Eq. (8), Eq. (9) are satisfied, with constant step-size $\gamma_k = \gamma_* \in \left(0, \frac{2}{3\sqrt{10}\,L}\right)$, and the choice of $\beta_k = \frac{2}{k+6}$.*

**Theorem 2.** [Proof in Appx. A.4.] *Suppose $G$ is monotone and $L$-Lipschitz, and let $x^\star$ satisfy $G(x^\star) = 0$. Consider the update of ($\square$). If the step-size $\gamma$ and anchoring coefficient $\beta_k$ satisfy the conditions of Lemma 2, then the Lyapunov function of Eq. (1) is decreasing. This implies the bound*

$$\|G(x_k)\|^2 \leq \frac{16/\gamma_*^2 + 32L^2}{(k+4)^2}\|x_0-x^\star\|^2. \qquad (10)$$

*Moreover, the constant $16/\gamma_*^2 + 32L^2$ is decreasing in $\gamma_*$, so it is smallest at the largest admissible step; as $\gamma_* \to \frac{2}{3\sqrt{10}\,L}$ it gives*

$$\|G(x_k)\|^2 \leq \frac{392L^2}{(k+4)^2}\|x_0-x^\star\|^2. \qquad (11)$$

**Summary** In the deterministic monotone Lipschitz setting, the optimal $O(1/k^2)$ last-iterate rate is already attained by several accelerated methods, including EAG (Yoon & Ryu, 2021), FEG (Lee & Kim, 2021), and anchored Popov (Tran-Dinh & Luo, 2021). GOMA matches this optimal rate under both schedules (fixed $\eta_*$ and fixed $\gamma_*$). Beyond matching the rate, our analysis yields a pseudo fixed-step-size scheme: hyperparameter tuning reduces to adjusting only $\eta_*$ or $\gamma_*$, with the two-time-scale structure maintained via the $(1-\beta_k)$ factor. This is simpler than the changing-step-size argument required by anchored Popov (Tran-Dinh & Luo, 2021) for a similar method. The main novelty of our work lies in the stochastic setting (Section 5).

## 5. Variant of GOMA for Stochastic Settings

### 5.1. Background

While first-order methods such as extragradient and optimistic gradient enjoy strong guarantees in deterministic variational inequalities, their behavior can fundamentally change in stochastic settings. In particular, under unbiased stochastic oracles, both extragradient and optimistic methods may fail to converge, even for simple monotone problems (Hsieh et al., 2020).

A line of research (Hsieh et al., 2019; Beznosikov et al., 2023) establishes convergence guarantees for *ergodic* (time-averaged) iterates of stochastic VI algorithms. However, ergodic convergence does not imply convergence of the actual iterates, which may cycle indefinitely despite having finite regret (Bailey et al., 2020).

This limitation has motivated approaches that modify the algorithmic dynamics to recover *last-iterate* convergence in stochastic settings. Hsieh et al. (2020) showed that introducing a two-time-scale scheme by using a larger step size for the extrapolation step restores almost sure *last-iterate* convergence of stochastic extragradient for affine problems.

Under general monotone and Lipschitz assumptions, strong last-iterate convergence guarantees in stochastic settings remain limited. Notably, Lee & Kim (2021) established last-iterate convergence of the Fast Extragradient method using a stochastic oracle with a suitably growing batch size. Cai et al. (2022a) incorporated recursive variance reduction into anchored Popov (Tran-Dinh & Luo, 2021) to obtain stochastic last-iterate convergence. More recently, Chen & Luo (2024) applied the same recursive variance reduction technique to an anchored extragradient-type method with a restarting mechanism, and established near-optimal stochastic last-iterate convergence guarantees for smooth convex–concave minimax problems.

These last-iterate guarantees rely on one of two mechanisms, each with its own cost. The first is growing batch sizes (Lee & Kim, 2021), which suppress the noise by drawing more samples each iteration. As a result, the per-iteration cost grows without bound and breaks the one-sample-per-round online model; in finite-sum problems it eventually reverts to full-batch gradients. By contrast, variance reduction (Cai et al., 2022a; Chen & Luo, 2024) keeps the batch fixed. To do so, however, it recomputes periodic large-batch snapshots and stores a reference gradient within multi-phase schedules. Moreover, its empirical advantage over plain stochastic methods has been reported to be limited for deep networks (Defazio & Bottou, 2019). We therefore ask whether last-iterate convergence is attainable with a single stochastic sample per iteration and constant, non-vanishing noise.

### 5.2. Method

We consider a simplified variant of (GOMA) in which $\gamma_k = 0$, leading to the following single–query update:

$$\begin{aligned} y_k &= \beta_k x_0 + (1-\beta_k)x_k, \\ x_{k+1} &= y_k - \eta_k G(y_k). \end{aligned} \qquad (\diamond)$$

This method evaluates the operator at an *anchored interpolation* between the current iterate and the initial point, and

then applies a single forward step at this interpolated point.

Unlike optimistic or extragradient-type methods, this variation of GOMA does not reuse past gradients, which removes a key source of instability under stochastic noise. The update can be interpreted as an extreme form of time-scale separation, where extrapolation is replaced by anchoring to a fixed reference point, yielding a stable single-query method in regimes where classical approaches fail.

Equation ($\diamond$) can also be written in the equivalent form

$$x_{k+1} = x_k + \beta_k(x_0 - x_k) - \eta_k G(x_k + \beta_k(x_0 - x_k)). \tag{12}$$

This formulation closely resembles Nesterov's momentum method (Nesterov, 1983):

$$x_{k+1} = x_k + \beta_k(x_k - x_{k-1}) - \eta_k G(x_k + \beta_k(x_k - x_{k-1})). \tag{13}$$

The key difference is that (12) anchors the extrapolation to the fixed point $x_0$, whereas Nesterov's method (13) anchors it to the previous iterate $x_{k-1}$. Moreover, the anchoring directions are opposite: replacing $x_0$ by $x_{k-1}$ in ($\diamond$) yields an update that matches Nesterov's momentum only when the momentum coefficient is negative.

An earlier work (Gidel et al., 2019a) showed the effectiveness of Polyak's heavy-ball method (Polyak, 1964) with negative momentum for game dynamics. We will further compare this approach with GOMA empirically, demonstrating the effectiveness of GOMA in stochastic settings.

We now turn to the convergence analysis of the stochastic variant ($\diamond$). In what follows, we state the assumptions under which last-iterate convergence can be established, and then present the corresponding rate guarantees.

**Assumption 3.** *We assume that $\widehat{G}(x, \xi)$ is an unbiased stochastic oracle for $G(x)$, i.e.,*

$$\mathbb{E}_\xi[\widehat{G}(x, \xi) \mid x] = G(x),$$

*and that, for some $\sigma \geq 0, \kappa > 0$, the noise satisfies the second moment conditional bound*

$$\mathbb{E}_\xi\left[\|\widehat{G}(x, \xi)\|^2 \mid x\right] \leq \sigma^2 + \kappa\|G(x)\|^2.$$

This assumption is similar to (Hsieh et al., 2019, Assump. 2) and holds under mild conditions. We take $\kappa \geq 1$, without loss of generality.[1] Crucially, our analysis covers any $\kappa \geq 1$,

---

[1] Indeed, by unbiasedness and the decomposition $\mathbb{E}_\xi[\|\widehat{G}(x, \xi)\|^2 \mid x] = \|G(x)\|^2 + \mathbb{E}_\xi[\|\widehat{G}(x, \xi) - G(x)\|^2 \mid x]$, Assumption 3 implies that $0 \leq \mathbb{E}_\xi\left[\|\widehat{G}(x, \xi) - G(x)\|^2 \mid x\right] \leq \sigma^2 + (\kappa - 1)\|G(x)\|^2$. If $\kappa < 1$, it follows that $\|G(x)\|^2 \leq \frac{\sigma^2}{1-\kappa}$ for all $x \in \mathbb{R}^d$, i.e., $G(x)$ is bounded. Furthermore, if Assumption 3 holds for some pair of constants $(\sigma, \kappa_0)$, then it also holds for every $\kappa \geq \kappa_0$ with the same $\sigma$. Hence, any admissible $\kappa_0 < 1$ can be replaced by $\kappa = 1$.

i.e. state-dependent noise whose second moment grows with $|G|^2$; prior single-call stochastic VI guarantees (E-Halpern, RAIN++) and FEG require bounded variance ($\kappa = 1$).

Under Assumption 3, we analyze ($\diamond$) with a stochastic oracle $\widehat{G}$ in place of $G$:

$$\begin{aligned} y_k &= \beta_k x_0 + (1 - \beta_k)x_k, \\ x_{k+1} &= y_k - \eta_k \widehat{G}(y_k, \xi_k). \end{aligned} \tag{14}$$

**Proof strategy.** Our analysis separates the deterministic and stochastic components of the dynamics. We compare the noisy iterates to a *deterministic reference trajectory*, obtained by running the same method ($\diamond$) with the exact operator and the same schedules:

$$\begin{aligned} \bar{x}_0 &= x_0, \qquad \bar{y}_k = \beta_k x_0 + (1 - \beta_k)\bar{x}_k, \\ \bar{x}_{k+1} &= \bar{y}_k - \eta_k G(\bar{y}_k). \end{aligned} \tag{15}$$

By $L$-Lipschitzness, $\|G(x_N)\|^2 \leq 2\|G(\bar{x}_N)\|^2 + 2L^2\|x_N - \bar{x}_N\|^2$, so it suffices to bound the residual along the reference trajectory and the mean-square deviation of the stochastic iterates from it. The first ingredient is purely deterministic: with the more conservative, $\kappa$-dependent step-size required by the stochastic setting, the noiseless method retains an $O(1/\sqrt{k})$ last-iterate guarantee, both at the iterates $\bar{x}_k$ and at the query points $\bar{y}_k$. Its proof is a Lyapunov analysis along the reference trajectory, analogous to the deterministic case.

**Lemma 3** (Deterministic reference bound). [Proof in Appx. B.2.] *Let $G$ be monotone and $L$-Lipschitz with $G(x^\star) = 0$, and let $(\bar{x}_k, \bar{y}_k)$ be given by (15) with $\beta_k = \frac{1}{k+2}$, $\eta_k = \frac{1}{L\sqrt{\kappa}(k+2)^{3/4}}$, and $\kappa \geq 1$. Then for all $N \geq 1$ and all $k \geq 0$,*

$$\|G(\bar{x}_N)\|^2 \leq \frac{33\, L^2 \kappa\, \|x_0 - x^\star\|^2}{\sqrt{N+1}}, \tag{16}$$

$$\|G(\bar{y}_k)\|^2 \leq \frac{94\, L^2 \kappa\, \|x_0 - x^\star\|^2}{\sqrt{k+1}}. \tag{17}$$

The bound (17) at the query points is the technically important addition: under Assumption 3 the second moment of the oracle grows with $\|G(y_k)\|^2$, so controlling the noise injected at step $k$ requires a residual bound along the *entire* trajectory, not only at the final iterate. The second ingredient shows that the noisy iterates track the reference trajectory.

**Lemma 4** (Stochastic stability). [Proof in Appx. B.3.] *In the setting of Lemma 3, let $\widehat{G}$ satisfy Assumption 3, let $(x_k)$ be given by (14), and set $e_k := x_k - \bar{x}_k$. Then for all $N \geq 0$,*

$$\mathbb{E}\|e_N\|^2 \leq \frac{1}{\sqrt{N+1}}\left(\frac{4\sigma^2}{L^2\kappa} + 752\,\kappa\,\|x_0 - x^\star\|^2\right). \tag{18}$$

*Table 1.* Last-iterate convergence guarantees on monotone Lipschitz VIs. Rates are reported on $\mathbb{E}\|G(x_k)\|$, i.e. the square root of the rates in Theorem 4, unless explicitly marked "(gap)", in which case they are on $\mathbb{E}[\text{GAP}]$ (not directly comparable to $\mathbb{E}\|G\|$ rates). "Unbounded domain" indicates the method's analysis applies on $\mathbb{R}^d$. "Single-call" = one operator evaluation per iteration.

| Method | Unbounded domain | Single-call | Deterministic rate | Stochastic rate | No variance reduction | No growing batch | Unbounded noise ($\kappa > 1$) |
|---|---|---|---|---|---|---|---|
| DSEG (Hsieh et al., 2020) | ✓ | ✗ | $O(k^{-1/2})$ | $O(k^{-1/2})$ (affine) | ✓ | ✓ | ✗ |
| FEG (Lee & Kim, 2021) | ✓ | ✗ | $O(k^{-1})$ | $O(k^{-1})$ | ✓ | ✗ | ✗ |
| E-Halpern (Cai et al., 2022a) | ✓ | ✓ | $O(k^{-1})$ | $O(k^{-1/3})$ | ✗ | ✓ | ✗ |
| RAIN$^{++}$ (Chen & Luo, 2024) | ✓ | ✗ | $O(k^{-1})$ | $\tilde{O}(k^{-1/2})$ | ✗ | ✓ | ✗ |
| GABPP (Abe et al., 2025) | ✗ | ✓ | $\tilde{O}(k^{-1})$ (gap) | $\tilde{O}(k^{-1/7})$ (gap) | ✓ | ✓ | ✗ |
| **GOMA simplified** ($\gamma_k = 0$) | ✓ | ✓ | $O(k^{-1/2})$ Thm 5 | $O(k^{-1/4})$ Thm 4 | ✓ | ✓ | ✓ |

The two terms in (18) mirror the two noise sources in Assumption 3: the additive variance $\sigma^2$ and the state-dependent part $\kappa\|G\|^2$, the latter controlled through (17). The key mechanism is that anchoring contracts the deviation: the interpolation toward $x_0$ multiplies the error by $(1 - \beta_k)$ at every step, which yields a contraction rate of $1 - \Theta(1/k)$ in the error recursion. This contraction is strong enough to keep the accumulated noise at $O(1/\sqrt{N})$ with a single sample per iteration, without variance reduction. Combining the two lemmas through the Lipschitz decomposition above yields our main stochastic guarantee.

**Theorem 4** (Last-iterate bound for stochastic GOMA). [Proof in Appx. B.4.] *Let $G : \mathbb{R}^d \to \mathbb{R}^d$ be monotone and $L$-Lipschitz and $\hat{G}(x, \xi)$ be a stochastic oracle following Assumption 3. Then for the updates described in (14) with $\beta_k = \frac{1}{k+2}$ and $\eta_k = \frac{1}{L\sqrt{\kappa}(k+2)^{3/4}}$, we have for all $N \geq 0$,*

$$\mathbb{E}\|G(x_N)\|^2 \leq \frac{1570 L^2 \kappa \|x_0 - x^\star\|^2}{\sqrt{N+1}} + \frac{8\sigma^2}{\kappa\sqrt{N+1}}.$$

Theorem 4 establishes, to the best of our knowledge, the first last-iterate convergence guarantee in the squared operator norm $\mathbb{E}\|G(x_T)\|^2$ for unconstrained stochastic monotone Lipschitz VIs, **without variance reduction or growing batch sizes** and the guarantee holds for every $\kappa \geq 1$, covering multiplicative noise. Concretely, GOMA attains $\mathbb{E}\|G(x_T)\|^2 \leq \varepsilon$ in $T = O(1/\varepsilon^2)$ iterations.

We now place our stochastic guarantees in context by comparing them to existing last-iterate results.

**Comparison with FEG.** FEG (Lee & Kim, 2021) requires two operator evaluations per iteration, misaligning it with the standard online-learning model. Its stochastic guarantee requires the per-iteration variance to decay as $\sigma_k = O(1/k)$; under constant noise the error term accumulates as $O(k)$, so the bound no longer vanishes and last-iterate convergence is lost. Achieving such decay requires growing minibatches. In contrast, our analysis (Theorem 4) guarantees convergence under Assumption 3 with a constant batch size and non-vanishing noise.

**Comparison with RAIN++.** Chen & Luo (2024) propose the RAIN/RAIN++ framework, which combines anchoring with recursive variance reduction to obtain near-optimal stochastic first-order oracle (SFO) complexity for smooth convex–concave minimax problems, matching the lower bound they derive up to logarithmic factors. Their algorithm relies on multi-phase schemes with restarting schedules and several hyperparameters that depend on problem constants.

In contrast, our analysis (Theorem 4) uses a single-call, single-phase update without variance reduction or restarts. Finally, we note that RAIN/RAIN++ do not formally return the last iterate, as they correspond to restarting SEG at an iterate uniformly sampled along the trajectory (hence not the last iterate most of the time).

**Comparison with E-Halpern.** Cai et al. (2022a) build on anchored Popov with recursive variance reduction (PAGE) to obtain a single-call algorithm with SFO complexity $O(1/\varepsilon^3)$ on $\mathbb{E}\|G\|$ in the monotone Lipschitz setting. Their analysis additionally requires Lipschitz continuity of the stochastic oracle in expectation, which we do not assume. Our analysis (Theorem 4) trades a slower SFO complexity of $O(1/\varepsilon^4)$ on $\mathbb{E}\|G\|$ for the removal of variance reduction and of the Lipschitz-in-expectation oracle assumption.

**Comparison with Abe et al. (2025).** Abe et al. (2025) propose GABPP, a single-call payoff-perturbed algorithm with periodic anchor restarts ($T_\sigma = O(T^{6/7})$), and prove $\mathbb{E}[\text{GAP}(\pi^{T+1})] = \tilde{O}(T^{-1/7})$ under bounded-variance noise (Theorem 4.3). Their setting differs structurally from ours: constrained monotone games on a compact $X$ with the gap function as progress measure, versus our unconstrained $\mathbb{R}^d$ with the squared operator norm (Section 2 explains why these measures are not interchangeable). They additionally require a uniform operator bound which is valid only on compact $X$; our state-dependent noise model handles unbounded $\|G\|$ directly. Within these complementary regimes, our rate is faster: $O(T^{-1/4})$ on $\mathbb{E}\|G(x_T)\|$, versus their $\tilde{O}(T^{-1/7})$ on $\mathbb{E}[\text{GAP}]$.

**Summary.** As Table 1 shows, methods that use grow-

ing batches or variance reduction achieve better stochastic rates because these mechanisms reduce or eliminate the noise terms that complicate stochastic last-iterate analysis. GOMA attains a last-iterate rate of $\mathcal{O}(1/\sqrt{k})$ on $\mathbb{E}\|G(x_k)\|^2$ (equivalently $\mathcal{O}(k^{-1/4})$ on $\mathbb{E}\|G(x_k)\|$), i.e. an SFO complexity of $\mathcal{O}(1/\varepsilon^4)$. This does not match the optimal rate of $\tilde{\mathcal{O}}(1/k)$ on $\mathbb{E}\|G(x_k)\|^2$ ($\tilde{\mathcal{O}}(1/\varepsilon^2)$ SFO complexity), which Chen & Luo (2024) establish as a lower bound and which variance-reduced, growing-batch methods attain. Nevertheless, GOMA delivers the best stochastic last-iterate rate on monotone Lipschitz VIs among methods that use neither variance reduction nor growing batches. Closing this gap remains an open question.

# 6. Experiments

## 6.1. Negative-Comonotone Quadratic Saddle Point (Deterministic)

**Setup.** We performed a toy experiment on a simple quadratic function also used in (Lee & Kim, 2021),

$$f(x,y) = -\tfrac{1}{6}x^2 + \tfrac{2\sqrt{2}}{3}xy + \tfrac{1}{6}y^2. \tag{19}$$

This instance is $\rho$-comonotone with $\rho = -1/3 < 0$ (i.e., *negative comonotone*), which lies outside the scope of our theory (our analysis requires monotonicity, $\rho \geq 0$). We include it for direct comparison with prior work (Lee & Kim, 2021), and provide an additional experiment on a *monotone* instance covered by our theory, which is in Appendix C.5.

**Methods.** We compare several first-order methods on this problem, including EG, DSEG, EAG-C, EAG-V, Nesterov, FEG, anchored Popov, and our proposed GOMA. For GOMA, we use a constant stepsize $\eta = 0.2$ and $\gamma_k = 0.8(1 - \beta_k)$ with $\beta_k = \frac{2}{k+6}$. All methods are evaluated by plotting the squared operator norm $\|F(z_k)\|^2$ against the number of gradient calls.

**Hyperparameter selection.** All baseline methods are run under the identical experimental setting of Figure 2 in (Lee & Kim, 2021), including the same initialization, step-size rules, and algorithmic parameters. This allows for a direct and fair comparison with the original results, to which we additionally include our proposed GOMA.

**Results.** See Figure 1. GOMA and FEG converge with an accelerated rate, whereas EG, DSEG, EAG-C, EAG-V, Anchored Popov and Nesterov diverge. Moreover, on this instance (under our tuning), GOMA yields uniformly smaller residuals than FEG by an approximately constant factor, with near-parallel log–log curves indicating the same asymptotic rate but a better constant.

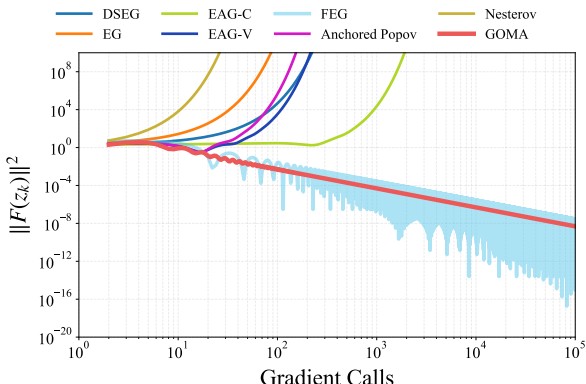

*Figure 1.* Quadratic experiment in the deterministic setting §6.1. Only GOMA and FEG converge. GOMA converges without oscillations, unlike FEG's dynamics.

## 6.2. Stochastic Games

We consider two stochastic settings. The first is a low-dimensional toy problem ($d = 2$) that satisfies the additive Gaussian noise assumption ($\kappa = 1$). The second is a finite-sum saddle-point problem ($d = 10$) with state-dependent multiplicative noise ($\kappa > 1$). Theorem 4 provides guarantees in both regimes, while the stochastic theory of FEG, E-Halpern, and RAIN++ applies only to the first.

**Methods.** We compare GOMA ($\diamond$) against DSEG, FEG, E-Halpern (with PAGE variance reduction), RAIN++, and Nesterov's accelerated method (with negative momentum, following (Gidel et al., 2019a)). GOMA is run with a **single stochastic sample per iteration**, matching the constant-batch setting of Theorem 4; we use no variance reduction and no growing batch size. Several of the baselines (E-Halpern, RAIN++) rely on variance reduction by design, and we use their authors' recommended implementations.

### 6.2.1. CASE I: BOUNDED VARIANCE ($\kappa = 1$)

**Setup.** We consider the stochastic bilinear game

$$f(x,y) = Lxy, \quad L = 1, \tag{20}$$

with saddle operator $F(z) = F(x,y) = (Ly, -Lx)$ and solution $z^\star = (0,0)$. The stochastic oracle is given by $\widehat{F}(z,\xi) = F(z) + \xi$, where $\xi \sim \mathcal{N}(0, \sigma^2 I)$ with $\sigma = 0.5$. Since the noise is state-independent, this setting satisfies Assumption 3 with $\kappa = 1$. We initialize at $z_0 = (1,1)$ and run all methods for $10^3$ gradient calls. The details about the hyperparameter selection are deferred to Appendix C.3.

**Results.** Figure 2 (left) compares the convergence of the operator norm under additive noise. **GOMA** achieves the fastest convergence, reaching a residual nearly an order of magnitude smaller than all baselines within $10^3$ gradient calls. **E-Halpern** exhibits steady progress due to its

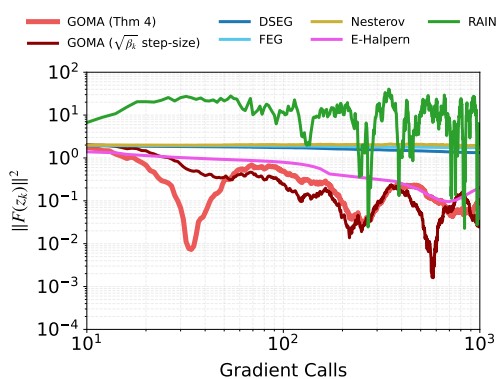 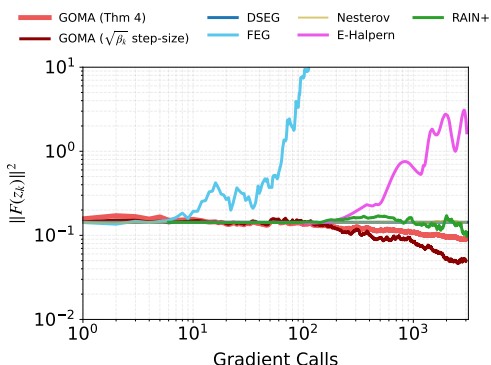

**Figure 2.** Stochastic experiments §6.2. Norm of the gradient operator vs. the number of gradient calls. **Left:** Stochastic bilinear example with additive noise ($\sigma = 0.5$), §6.2.1. **Right:** Finite-sum saddle-point problem with multiplicative noise, §6.2.2. **Stochastic GOMA consistently outperforms the baselines in both tasks**, despite using a milder hyperparameter search and simpler setup.

variance-reduction mechanism; however, it converges more slowly and plateaus around $10^{-1}$.

The two-time-scale extragradient-type methods, DSEG and FEG, show little to no convergence, remaining close to their initial values throughout the experiment. RAIN displays highly unstable behavior with large oscillations and fails to converge. These results highlight the effectiveness of combining anchoring with single-call stochastic gradients: **GOMA achieves the best convergence without relying on variance reduction or multi-point oracle evaluations**.

6.2.2. CASE II: STATE-DEPENDENT VARIANCE ($\kappa > 1$)

**Setup.** We consider the finite-sum saddle problem

$$\min_{\theta \in \mathbb{R}^d} \max_{\varphi \in \mathbb{R}^d} \frac{1}{n} \sum_{i=1}^{n} \left( \theta^\top b_i + \theta^\top A_i \varphi + c_i^\top \varphi \right),$$

with saddle operator $F(\theta, \varphi) = [\bar{b} + \bar{A}\varphi; -(\bar{A}^\top \theta + \bar{c})]$, where $\bar{A} = \frac{1}{n} \sum_i A_i$ and $\bar{b}, \bar{c}$ are the sample means. We set $n = d = 10$ and $A_i = \mathrm{diag}(0, \ldots, \lambda_i, \ldots, 0)$ with $\lambda_i$ equally spaced in $[\tau, 1]$, $\tau = 0.1$. The SFO returns $F_i(\theta, \varphi) = [b_i + A_i\varphi; -(A_i^\top \theta + c_i)]$, sampling $i$ uniformly.

**Hyperparameter selection.** For GOMA, we use $\beta_k = 1/(k+2)$ and $\eta_k = c\sqrt{\beta_k}$ with $c$ selected via grid search. Full details are provided in Appendix C.4.

**Results.** Figure 2 (right) illustrates the convergence behavior of the operator norm under multiplicative noise. This setting is covered by Theorem 4 but falls outside the bounded-variance assumptions of FEG, E-Halpern, and RAIN++. Both RAIN++ and GOMA exhibit convergence empirically.

In contrast, DSEG stagnates at a high plateau (around $10^{-1}$) and fails to make further progress. As shown in Appx. C.2, its theory predicts arbitrarily slow convergence in higher dimensions, which is consistent with this behavior. Moreover, FEG and E-Halpern diverge.

Overall, these results show that **GOMA converges in this multiplicative noise regime as guaranteed by Theorem 4, while the bounded-variance baselines (FEG, E-Halpern) exit their theoretical regime and diverge**.

## 7. Discussion

Our results show that anchoring, when combined with generalized optimistic dynamics, offers a principled approach to addressing three central challenges in variational inequality algorithms: per-iteration efficiency, robustness to stochasticity, and last-iterate acceleration.

In particular, GOMA attains the optimal $O(1/k^2)$ last-iterate rate in deterministic monotone Lipschitz settings using only a single gradient evaluation per iteration, applicable for online learning regimes, in contrast to extragradient-based methods. Moreover, the stochastic variant of GOMA achieves a $O(1/\sqrt{k})$ last-iterate convergence rate.

Compared to existing baselines, both the algorithm and its analysis work with minimal assumptions of problem parameters such as noise level or strong monotonicity. Consequently, GOMA consistently performs best across our stochastic experiments, underscoring its robustness for stochastic optimization and variational inequality problems.

The most important open direction for the stochastic theory is to close the gap to the SFO lower bound established by Chen & Luo (2024) without resorting to variance reduction or growing batches. Last-iterate convergence of GOMA in the constrained setting, where the convergence measure changes from $\|G\|^2$ to the gap function, remains open. Beyond the monotone setting, extending the analysis to broader operator classes such as negative comonotone operators is another natural direction. Finally, applying these methods at scale in reinforcement learning and adversarial training, where stochasticity, stability, and gradient efficiency are central concerns, is an exciting practical direction.

## Impact Statement

This paper presents work aimed at advancing the field of Variational Inequality Problems. Our work being focused on the theoretical aspect, we do not foresee any direct societal impact. Regarding indirect impact, while a common positive, foreseeable impact of such research aiming to find better optimisation algorithms is the more efficient use of computing resources, it is important to be mindful that, across history, cost-lowering technological improvements have nevertheless often led to an increase in consumption due to the Jevons paradox.

## Acknowledgements

This research was partially supported by the Canada CIFAR AI Chair program (Mila), Simon Lacoste-Julien is a CIFAR Associate Fellow in the Learning in Machines & Brains program.

We would like to thank Zichu Liu, Juan David Guerra and Mehran Shakerinava for their feedbacks on the initial draft of this paper.

We also acknowledge the use of AI assistants during this work. ChatGPT (OpenAI) and Claude Code (Anthropic) helped identify errors in intermediate steps while we were developing the proofs of our main theorems. In particular, ChatGPT (OpenAI, Pro mode) found a sign error that invalidated an earlier version of our stochastic analysis and proposed the corrected proof strategy for Theorem 4, based on a deterministic reference trajectory and a stochastic stability argument.

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

# Appendix

## A. Proof of Section 4

**Generalized Optimistic Method with Anchoring (GOMA)**

$$
\begin{aligned}
y_k &= \beta_k x_0 + (1 - \beta_k)x_k - \gamma_k G(y_{k-1}) \\
x_{k+1} &= \beta_k x_0 + (1 - \beta_k)x_k - \eta_k G(y_k),
\end{aligned}
\tag{GOMA}
$$

Define the Lyapunov function

$$
V_k = a_k \|G(x_k)\|^2 + b_k \langle G(x_k),\, x_k - x_0 \rangle + c_k L^2 \|x_k - y_{k-1}\|^2.
\tag{1}
$$

Where $a_k = c_k = \frac{b_k \eta_k}{2\beta_k}$, and $b_{k+1} = \frac{b_k}{1-\beta_k}$.

### A.1. Proof of Lemma 1

This proof is inspired from the proof of anchored Popov in (Tran-Dinh & Luo, 2021).

**Lemma 1.** *Let $G$ be monotone and $L$-Lipschitz, and consider the iterates*

$$
\begin{aligned}
y_k &= \beta_k x_0 + (1 - \beta_k)x_k - \eta_*(1 - \beta_k)G(y_{k-1}), \\
x_{k+1} &= \beta_k x_0 + (1 - \beta_k)x_k - \eta_* G(y_k).
\end{aligned}
\tag{$\triangle$}
$$

*We prove that the Lyapunov function (1) is decreasing if the step-size $\eta_k$ satisfies the following conditions:*

$$
\eta_{k+1} \;\leq\; \frac{\beta_{k+1}}{2M\,\eta_k\,\beta_k(1-\beta_k)}.
\tag{2}
$$

$$
1 - 2M\eta_k^2(1-\beta_k)^2 - M\eta_k^2\beta_k^2 \geq 0.
\tag{3}
$$

$$
\eta_{k+1} \leq \frac{\beta_{k+1}}{2\,\beta_k(1-\beta_k)}\left[\frac{2(1-\beta_k^2) - 4M\eta_k^2(1-\beta_k)^2 - \beta_k^2}{1 - 2M\eta_k^2(1-\beta_k)^2 - M\eta_k^2\beta_k^2}\eta_k\right].
\tag{4}
$$

*for $M := 2L^2(1 + \theta)$ and $\theta \geq 0$. With $\widetilde{c}_{k+1} \geq 0$ for $\theta = 2$, these give:*

$$
V_k - V_{k+1} \geq \widetilde{c}_{k+1}L^2\|x_{k+1} - y_k\|^2.
$$

*Proof.* First, from the update equations we obtain the three key difference identities:

$$
x_{k+1} - x_k = \beta_k(x_0 - x_k) - \eta_k G(y_k),
\tag{21}
$$

$$
x_{k+1} - x_k = \frac{\beta_k}{1 - \beta_k}(x_0 - x_{k+1}) - \frac{\eta_k}{1 - \beta_k}G(y_k),
\tag{22}
$$

$$
x_{k+1} - y_k = -\eta_k\big[G(y_k) - (1 - \beta_k)G(y_{k-1})\big].
\tag{23}
$$

Next, monotonicity of $G$ gives

$$
\langle G(x_{k+1}) - G(x_k),\, x_{k+1} - x_k \rangle \geq 0
$$

$$
\langle G(x_{k+1}),\, x_{k+1} - x_k \rangle \geq \langle G(x_k),\, x_{k+1} - x_k \rangle
$$

Using Eq. (21) and Eq. (22), we write:

$$\langle G(x_{k+1}), \frac{\beta_k}{1-\beta_k}(x_0 - x_{k+1}) - \frac{\eta_k}{1-\beta_k}G(y_k) \rangle \geq \langle G(x_k), \beta_k(x_0 - x_k) - \eta_k G(y_k) \rangle$$

Rearranging:

$$\frac{\beta_k}{1-\beta_k}\langle G(x_{k+1}), x_0 - x_{k+1} \rangle \geq \beta_k \langle G(x_k), x_0 - x_k \rangle - \eta_k \langle G(x_k), G(y_k) \rangle + \frac{\eta_k}{1-\beta_k}\langle G(x_{k+1}), G(y_k) \rangle$$

Multiplying this inequality by $\frac{b_k}{\beta_k}$ and taking $b_{k+1} = \frac{b_k}{1-\beta_k}$:

$$\underbrace{b_k \langle G(x_k), x_k - x_0 \rangle - b_{k+1}\langle G(x_{k+1}), x_{k+1} - x_0 \rangle}_{T[1]} \geq \frac{b_k \eta_k}{\beta_k(1-\beta_k)}\langle G(x_{k+1}), G(y_k) \rangle - \frac{b_k \eta_k}{\beta_k}\langle G(x_k), G(y_k) \rangle.$$

$$= b_{k+1}\eta_k \langle G(x_{k+1}), G(y_k) \rangle + \frac{b_k \eta_k}{\beta_k}\langle G(x_{k+1}) - G(x_k), G(y_k) \rangle.$$

Adding the $a_k$-terms and the $c_k$-terms gives

$$\begin{aligned} V_k - V_{k+1} \geq & a_k \|G(x_k)\|^2 - a_{k+1}\|G(x_{k+1})\|^2 + T[1] + c_k L^2 \|x_k - y_{k-1}\|^2 - c_{k+1}L^2 \|x_{k+1} - y_k\|^2 \\ \geq & a_k \|G(x_k)\|^2 - a_{k+1}\|G(x_{k+1})\|^2 \\ & + b_{k+1}\eta_k \langle G(x_{k+1}), G(y_k) \rangle + \frac{b_k \eta_k}{\beta_k}\langle G(x_{k+1}) - G(x_k), G(y_k) \rangle \\ & + c_k L^2 \|x_k - y_{k-1}\|^2 - c_{k+1}L^2 \|x_{k+1} - y_k\|^2 \end{aligned} \tag{24}$$

Next, we upper bound $\|G(y_k) - G(y_{k-1})\|^2$ as follow:

$$\begin{aligned} \|G(y_k) - G(y_{k-1})\|^2 &= \|G(y_k) - G(x_k) + G(x_k) - G(y_{k-1})\|^2 \\ &\leq 2\|G(x_k) - G(y_k)\|^2 + 2\|G(x_k) - G(y_{k-1})\|^2 \\ &\leq 2\|G(x_k)\|^2 - 4\langle G(x_k), G(y_k) \rangle + 2\|G(y_k)\|^2 + 2L^2 \|x_k - y_{k-1}\|^2. \end{aligned} \tag{25}$$

Where we used the Lipschitz inequality between $x_k$ and $y_{k-1}$ in the last inequality.

We consider the following and use Lipschitzness between $x_{k+1}$ and $y_k$ and Eq. (23) to bound the left-hand side

$$\|G(x_{k+1}) - G(y_k)\|^2 + \theta L^2 \|x_{k+1} - y_k\|^2 \leq (1 + \theta)L^2 \|x_{k+1} - y_k\|^2 \tag{26}$$

$$\leq (1 + \theta)L^2 \eta_k^2 \|G(y_k) - (1 - \beta_k)G(y_{k-1})\|^2 \tag{27}$$

We can upper bound the right-hand side. We first use the inequality $\|a + b\|^2 \leq 2\|a\|^2 + 2\|b\|^2$. Then we use the bound in the upper bound on $\|G(y_k) - G(y_{k-1})\|^2$.

$$\begin{aligned} & (1 + \theta)L^2 \eta_k^2 \|G(y_k) - (1 - \beta_k)G(y_{k-1})\|^2 \\ & \leq 2L^2 (1 + \theta)(1 - \beta_k)^2 \eta_k^2 \|G(y_k) - G(y_{k-1})\|^2 + 2L^2 (1 + \theta)\eta_k^2 \beta_k^2 \|G(y_k)\|^2 \\ & \overset{(25)}{\leq} 4L^2 (1 + \theta)\eta_k^2 (1 - \beta_k)^2 (\|G(x_k)\|^2 - 2\langle G(x_k), G(y_k) \rangle + \|G(y_k)\|^2) \\ & + 4L^4 (1 + \theta)\eta_k^2 (1 - \beta_k)^2 \|x_k - y_{k-1}\|^2 + 2L^2 (1 + \theta)\eta_k^2 \beta_k^2 \|G(y_k)\|^2 \end{aligned} \tag{28}$$

Now we expand the quadratic in the left-hand side of Eq. (27) and also substitute the right-hand side from above.

$$
\begin{aligned}
& \|G(x_{k+1})\|^2 - 2\langle G(x_{k+1}), G(y_k)\rangle + \|G(y_k)\|^2 + \theta L^2\|x_{k+1} - y_k\|^2 \\
& \leq 4L^2(1+\theta)\,\eta_k^2(1-\beta_k)^2\left(\|G(x_k)\|^2 - 2\langle G(x_k), G(y_k)\rangle + \|G(y_k)\|^2\right) \\
& + 4L^4(1+\theta)\,\eta_k^2(1-\beta_k)^2\,\|x_k - y_{k-1}\|^2 + 2L^2(1+\theta)\,\eta_k^2\,\beta_k^2\|G(y_k)\|^2
\end{aligned}
$$

Rearranging and setting $M := 2L^2(1+\theta)$, we get:

$$
\begin{aligned}
& \|G(x_{k+1})\|^2 - 2\langle G(x_{k+1}), G(y_k)\rangle - 2M\eta_k^2(1-\beta_k)^2\,\|G(x_k)\|^2 + 4M\eta_k^2(1-\beta_k)^2\,\langle G(x_k), G(y_k)\rangle \\
& + \left[1 - 2M\eta_k^2(1-\beta_k)^2 - M\eta_k^2\beta_k^2\right]\|G(y_k)\|^2 + \theta L^2\,\|x_{k+1} - y_k\|^2 \\
& - 2L^2 M\eta_k^2(1-\beta_k)^2\,\|x_k - y_{k-1}\|^2 \ \leq\ 0.
\end{aligned}
$$

Combine terms to get $G(x_{k+1}) - G(x_k)$:

$$
\begin{aligned}
& \|G(x_{k+1})\|^2 - 2(1 - 2M\eta_k^2(1-\beta_k)^2)\langle G(x_{k+1}), G(y_k)\rangle - 2M\eta_k^2(1-\beta_k)^2\,\|G(x_k)\|^2 \\
& - 4M\eta_k^2(1-\beta_k)^2\,\langle G(x_{k+1}) - G(x_k), G(y_k)\rangle + \left[1 - 2M\eta_k^2(1-\beta_k)^2 - M\eta_k^2\beta_k^2\right]\|G(y_k)\|^2 \\
& + \theta L^2\,\|x_{k+1} - y_k\|^2 - 2L^2 M\eta_k^2(1-\beta_k)^2\,\|x_k - y_{k-1}\|^2 \ \leq\ 0.
\end{aligned}
$$

We multiply this equation by $\frac{a_k}{2M\eta_k^2(1-\beta_k)^2}$ and add it to right-hand side of Eq. (24) get:

$$
\begin{aligned}
V_k - V_{k+1} \ \geq\ & \underbrace{\left(\frac{a_k}{2M\eta_k^2(1-\beta_k)^2} - a_{k+1}\right)}_{S_k^{11}} \|G(x_{k+1})\|^2 \\
& + 2\underbrace{\left(-\frac{a_k(1 - 2M\eta_k^2(1-\beta_k)^2)}{2M\eta_k^2(1-\beta_k)^2} + \frac{b_{k+1}\eta_k}{2}\right)}_{S_k^{12}} \langle G(x_{k+1}), G(y_k)\rangle \\
& + \underbrace{\left(\frac{a_k}{2M\eta_k^2(1-\beta_k)^2}(1 - 2M\eta_k^2(1-\beta_k)^2 - M\eta_k^2\beta_k^2)\right)}_{S_k^{22}} \|G(y_k)\|^2 \\
& + \underbrace{\left(-2a_k + \frac{b_k\eta_k}{\beta_k}\right)}_{S_k^{23}} \langle G(x_{k+1}) - G(x_k), G(y_k)\rangle \\
& + L^2\underbrace{\left(c_k - a_k\right)}_{\tilde{c}_k} \|x_k - y_{k-1}\|^2 \\
& + L^2\underbrace{\left(\frac{a_k\theta}{2M\eta_k^2(1-\beta_k)^2} - c_{k+1}\right)}_{\tilde{c}_{k+1}} \|x_{k+1} - y_k\|^2.
\end{aligned}
\qquad (29)
$$

We set $a_k = \frac{b_k\eta_k}{2\beta_k}$ and get $S_k^{23} = 0$. Also we set $c_k = a_k$, then $\tilde{c}_k = 0$.

Now, in order to prove the right-hand side is greater than zero, it is sufficient to show that $\tilde{c}_{k+1} \geq 0$, and $S_k \succeq 0$, where:

$$
S_k \ = \ \begin{pmatrix} S_k^{11} & S_k^{12} \\ S_k^{12} & S_k^{22} \end{pmatrix}.
$$

We simplify $S_k^{ij}$ further using that $a_k = \frac{b_k \eta_k}{2\beta_k}$ and $b_{k+1} = \frac{b_k}{1-\beta_k}$:

$$S_k^{11} := \frac{a_k}{2M\eta_k^2(1-\beta_k)^2} - a_{k+1} \qquad = \frac{b_k}{4M\,\eta_k\,\beta_k(1-\beta_k)^2} - \frac{b_k\eta_{k+1}}{2\beta_{k+1}\,(1-\beta_k)}, \tag{30}$$

$$S_k^{12} := -\frac{a_k(1-2M\eta_k^2(1-\beta_k)^2)}{2M\eta_k^2(1-\beta_k)^2} + \frac{b_{k+1}\eta_k}{2} \qquad = -\frac{b_k\left(1-2M\eta_k^2(1-\beta_k)\right)}{4M\,\eta_k\beta_k\,(1-\beta_k)^2}, \tag{31}$$

$$S_k^{22} := \frac{a_k}{2M\eta_k^2(1-\beta_k)^2}(1-2M\eta_k^2(1-\beta_k)^2 - M\eta_k^2\beta_k^2) \qquad = \frac{b_k}{4M\eta_k\beta_k(1-\beta_k)^2}(1-2M\eta_k^2(1-\beta_k)^2 - M\eta_k^2\beta_k^2). \tag{32}$$

We need $S_k^{11} \geq 0$, $S_k^{22} \geq 0$, and $S_k^{11}\, S_k^{22} \geq (S_k^{12})^2$.

$$S_k^{11} \geq 0 \iff \frac{b_k}{4M\,\eta_k\,\beta_k(1-\beta_k)^2} - \frac{b_k\eta_{k+1}}{2\beta_{k+1}\,(1-\beta_k)} \geq 0 \iff \eta_{k+1} \leq \frac{\beta_{k+1}}{2M\,\eta_k\,\beta_k(1-\beta_k)}, \tag{33}$$

$$S_k^{22} \geq 0 \iff 1-2M\eta_k^2(1-\beta_k)^2 - M\eta_k^2\beta_k^2 \geq 0, \iff \eta_k \leq \frac{1}{\sqrt{M[2(1-\beta_k)^2 + \beta_k^2]}} \tag{34}$$

$S_k^{11}S_k^{22} \geq (S_k^{12})^2 \iff$

$$\left(\frac{b_k}{4M\,\eta_k\,\beta_k(1-\beta_k)^2} - \frac{b_k\eta_{k+1}}{2\beta_{k+1}\,(1-\beta_k)}\right)\frac{b_k}{4M\eta_k\beta_k(1-\beta_k)^2}(1-2M\eta_k^2(1-\beta_k)^2 - M\eta_k^2\beta_k^2)$$

$$\geq \left(\frac{b_k\left(1-2M\eta_k^2(1-\beta_k)\right)}{4M\,\eta_k\beta_k\,(1-\beta_k)^2}\right)^2$$

$$\iff \left(1 - \frac{2M\,\eta_k\,\beta_k(1-\beta_k)}{\beta_{k+1}}\eta_{k+1}\right)(1-2M\eta_k^2(1-\beta_k)^2 - M\eta_k^2\beta_k^2)$$

$$\geq \left(1-2M\eta_k^2(1-\beta_k)\right)^2$$

$$\iff 1 - \frac{\left(1-2M\eta_k^2(1-\beta_k)\right)^2}{1-2M\eta_k^2(1-\beta_k)^2 - M\eta_k^2\beta_k^2} \geq \frac{2M\,\eta_k\,\beta_k(1-\beta_k)}{\beta_{k+1}}\eta_{k+1}$$

$$\iff \frac{\beta_{k+1}}{2M\,\eta_k\,\beta_k(1-\beta_k)}\left(1 - \frac{\left(1-2M\eta_k^2(1-\beta_k)\right)^2}{1-2M\eta_k^2(1-\beta_k)^2 - M\eta_k^2\beta_k^2}\right) \geq \eta_{k+1}. \tag{35}$$

We further simplify Eq. (35):

$$\eta_{k+1} \leq \frac{\beta_{k+1}}{2M\,\eta_k\,\beta_k(1-\beta_k)}\left[\frac{2M\eta_k^2(1-\beta_k^2) - 4M^2\eta_k^4(1-\beta_k)^2 - M\eta_k^2\beta_k^2}{1-2M\eta_k^2(1-\beta_k)^2 - M\eta_k^2\beta_k^2}\right]$$

$$= \frac{\beta_{k+1}}{2\beta_k(1-\beta_k)}\left[\frac{2(1-\beta_k^2) - 4M\eta_k^2(1-\beta_k)^2 - \beta_k^2}{1-2M\eta_k^2(1-\beta_k)^2 - M\eta_k^2\beta_k^2}\right]\eta_k \tag{36}$$

If all these three condition hold, we will have:

$$V_k - V_{k+1} \geq L^2\left(\frac{a_k\theta}{2M\eta^2} - a_{k+1}\right)\|x_{k+1} - y_k\|^2.$$

We show the positivity of the term on the right-hand side, along with the three conditions in Lemma 1.

Next, we show that conditions of Eq. (2), Eq. (3), Eq. (4) are satisfied, with constant step-size $\eta_k = \eta_* \in (0, \frac{1}{2\sqrt{3}L})$, and the choice of $\beta_k = \frac{2}{k+6}$.

Let $\eta_k \equiv \eta \in \left(0, \frac{1}{\sqrt{2M}}\right)$ and $\beta_k = \frac{2}{k+6}$.

**Condition (3).** We compute

$$1 - 2M\eta^2(1 - \beta_k)^2 - M\eta^2\beta_k^2 = 1 - M\eta^2\Big(2(1 - \beta_k)^2 + \beta_k^2\Big).$$

For $\beta_k \in [0, 1]$, we have $2(1 - \beta_k)^2 + \beta_k^2 \in [3/4, 2]$. Thus

$$1 - M\eta^2\big(2(1 - \beta)^2 + \beta^2\big) \geq 1 - 2M\eta^2 > 0 \quad \text{since } M\eta^2 < \tfrac{1}{2}.$$

Hence (3) holds.

**Condition (2).** With $\eta_{k+1} = \eta_k = \eta$, condition (2) reads

$$2M\eta^2 \leq \frac{\beta_{k+1}}{\beta_k(1 - \beta_k)}.$$

For $\beta_k = \frac{2}{k+6}$ we compute

$$\frac{\beta_{k+1}}{\beta_k(1 - \beta_k)} = \frac{\frac{2}{k+7}}{\frac{2}{k+6}\left(1 - \frac{2}{k+6}\right)} = \frac{(k+6)^2}{(k+7)(k+4)} \geq 1.$$

Thus $2M\eta^2 \leq 1$, i.e. $\eta^2 \leq \frac{1}{2M}$, suffices. This is satisfied since $\eta < 1/\sqrt{2M}$.

**Condition (4).** With $\eta_{k+1} = \eta_k = \eta$ and $t := M\eta^2$, condition (4) reduces to

$$1 \leq \frac{\beta_{k+1}}{2\beta_k(1 - \beta_k)} \frac{2(1 - \beta_k^2) - 4t(1 - \beta_k)^2 - \beta_k^2}{1 - 2t(1 - \beta_k)^2 - t\beta_k^2}.$$

For $\beta_k = \frac{2}{k+6}$, the prefactor simplifies to

$$\frac{\beta_{k+1}}{2\beta_k(1 - \beta_k)} = \frac{(k+6)^2}{2(k+7)(k+4)} = \frac{1}{(2 + \beta_k)(1 - \beta_k)}.$$

Setting $\beta := \beta_k$, the inequality is equivalent to

$$\frac{2 - 3\beta^2 - 4t(1 - \beta)^2}{1 - 2t(1 - \beta)^2 - t\beta^2} \geq (2 + \beta)(1 - \beta). \tag{$\star$}$$

Define

$$F(t) := \frac{2 - 3\beta^2 - 4t(1 - \beta)^2}{1 - 2t(1 - \beta)^2 - t\beta^2} - (2 + \beta)(1 - \beta).$$

A derivative check shows $\partial_t F(t) < 0$ on $t \in (0, \frac{1}{2})$, so the worst case is at $t = \frac{1}{2}$. Plugging in $t = \frac{1}{2}$, we get

$$\frac{4\beta - 5\beta^2}{2\beta - \frac{3}{2}\beta^2} = \frac{4 - 5\beta}{2 - \frac{3}{2}\beta} \geq (2 + \beta)(1 - \beta) = 2 - \beta - \beta^2.$$

This inequality is equivalent to

$$0 \geq -\tfrac{1}{2}\beta^2 + \tfrac{3}{2}\beta^3 = \tfrac{1}{2}\beta^2(-1 + 3\beta),$$

which holds for $\beta \leq \frac{1}{3}$. Since $\beta_k \leq \frac{1}{3}$, condition (4) follows.

**Positivity of the Lyapunov decrease.** Given that $c_k = a_k$ we have:

$$V_k - V_{k+1} \geq L^2\left(\frac{a_k \theta}{2M\eta^2} - a_{k+1}\right) \|x_{k+1} - y_k\|^2.$$

Hence the first coefficient equals $a_k \beta_k^2 > 0$. For the second coefficient we use the schedule $a_k = \frac{b_k \eta}{2\beta_k}$ with $b_{k+1} = \frac{b_k}{1-\beta_k}$ (and constant $\eta$), which yields

$$\frac{a_{k+1}}{a_k} = \frac{b_{k+1}}{b_k}\frac{\beta_k}{\beta_{k+1}} = \frac{\beta_k}{\beta_{k+1}(1-\beta_k)}.$$

Therefore

$$\frac{a_k \theta}{2M\eta^2} - a_{k+1} \geq 0 \qquad \Longleftrightarrow \qquad 2M\eta^2 \leq \theta\frac{\beta_{k+1}(1-\beta_k)}{\beta_k}.$$

With the specific choice $\beta_k = \frac{2}{k+6}$ we have

$$\frac{\beta_{k+1}(1-\beta_k)}{\beta_k} = \frac{\frac{2}{k+7}\left(1 - \frac{2}{k+6}\right)}{\frac{2}{k+6}} = \frac{k+4}{k+7},$$

which takes its minimum at $k = 0$ and hence a sufficient condition is

$$2M\eta^2 \leq \frac{4\theta}{7}.$$

Since we know that $2M\eta^2 \leq 1$, we can be sure that with the choice of $\theta = 2$, this condition is satisfied.

With this choice of $\theta$, we have $M = 6L^2$. Therefore, the admissible range of $\eta$ is $\eta \in \left(0, \frac{1}{2\sqrt{3}L}\right)$, as stated in the lemma.

$\square$

### A.2. Proof of Theorem 1

**Theorem 1.** *Suppose $G$ is monotone and $L$-Lipschitz continuous. Consider the updates of Eq. ($\triangle$) With the parameter choices*

$$\beta_k = \frac{2}{k+6}, \quad \eta_* \in \left(0, \frac{1}{2\sqrt{3}L}\right),$$

$$a_k = c_k = \frac{b_0\,\eta_*}{80}(k+4)(k+5)(k+6),$$

$$b_k = \frac{b_0}{20}(k+4)(k+5).$$

*the Lyapunov decrease from Lemma 1 implies the bound*

$$\|G(x_k)\|^2 \leq \frac{16/\eta_*^2 + 72L^2}{(k+6)^2}\|x_0 - x^\star\|^2. \tag{5}$$

*Also, since the constant is smallest at the largest admissible step, as $\eta_* \to \frac{1}{2\sqrt{3}L}$ we get the bound:*

$$\|G(x_k)\|^2 \leq \frac{264L^2}{(k+6)^2}\|x_0 - x^\star\|^2.$$

*Proof.* Let $H_k := a_k\|G(x_k)\|^2 + b_k\langle G(x_k), x_k - x_0\rangle$. Then using the Young's inequality $\langle a, b\rangle \leq \frac{\alpha}{2}\|a\|^2 + \frac{1}{2\alpha}\|b\|^2$ with $\alpha = a_k$, we have

$$H_k = a_k\|G(x_k)\|^2 + b_k\langle G(x_k) - G(x^\star), x_k - x^\star\rangle + b_k\langle G(x_k), x^\star - x_0\rangle$$

$$\geq a_k\|G(x_k)\|^2 - \frac{a_k}{2}\|G(x_k)\|^2 - \frac{b_k^2}{2a_k}\|x_0 - x^\star\|^2$$

$$= \frac{a_k}{2}\|G(x_k)\|^2 - \frac{b_k^2}{2a_k}\|x_0 - x^\star\|^2. \tag{37}$$

Finally, from Eq. (37), we have

$$\frac{a_k}{2}\|G(x_k)\|^2 \le H_k + \frac{b_k^2}{2a_k}\|x_0 - x^\star\|^2,$$

leading to

$$\frac{a_k}{2}\|G(x_k)\|^2 + c_k L^2\|x_k - y_{k-1}\|^2 \le V_k + \frac{b_k^2}{2a_k}\|x_0 - x^\star\|^2. \tag{38}$$

We replace the values of $a_k, b_k, c_k$. Using $y_{-1} = x_0$, the estimate in Eq. (38), by induction, we can show that

$$\frac{(k+4)(k+5)(k+6)\,\eta_* \, b_0}{160}\|G(x_k)\|^2 + \frac{L^2(k+4)(k+5)(k+6)\,\eta_* \, b_0}{80}\|x_k - y_{k-1}\|^2$$

$$\le V_k + \frac{b_k^2}{2a_k}\|x_0 - x^\star\|^2$$

$$\le V_0 + b_0 \frac{(k+4)(k+5)}{10\eta_*(k+6)}\|x_0 - x^\star\|^2$$

$$= \frac{3b_0\eta_*}{2}\|G(x_0)\|^2 + b_0 \frac{(k+4)(k+5)}{10\eta_*(k+6)}\|x_0 - x^\star\|^2$$

$$\le \frac{3b_0\eta_* L^2}{2}\|x_0 - x^\star\|^2 + b_0 \frac{(k+4)(k+5)}{10\eta_*(k+6)}\|x_0 - x^\star\|^2$$

Multiplying by $\dfrac{160}{b_0\eta_*(k+4)(k+5)(k+6)}$ and noting that $(k+4)(k+5)(k+6) \ge \frac{5}{9}(k+6)^3$ (with equality at $k=0$), so that $\frac{1}{(k+4)(k+5)(k+6)} \le \frac{9}{5} \cdot \frac{1}{(k+6)^3} \le \frac{9}{5} \cdot \frac{1}{6} \cdot \frac{1}{(k+6)^2} = \frac{3}{10} \cdot \frac{1}{(k+6)^2}$, gives:

$$\|G(x_k)\|^2 \le \frac{240\,L^2}{(k+4)(k+5)(k+6)}\|x_0 - x^\star\|^2 + \frac{16}{\eta_*^2(k+6)^2}\|x_0 - x^\star\|^2$$

$$\le \frac{16/\eta_*^2 + 72L^2}{(k+6)^2}\|x_0 - x^\star\|^2.$$

Also if we pick the largest admissible constant stepsize $\eta_* = \frac{1}{2\sqrt{3}L}$ we get the bound:

$$\|G(x_k)\|^2 \le \frac{264L^2}{(k+6)^2}\|x_0 - x^\star\|^2.$$

$\square$

## A.3. Proof of Lemma 2

**Lemma 2.** *Let $G$ be monotone and L-Lipschitz, and consider the iterates*

$$\begin{aligned}y_k &= \beta_k x_0 + (1-\beta_k)x_k - \gamma_* \, G(y_{k-1}),\\ x_{k+1} &= \beta_k x_0 + (1-\beta_k)x_k - (1-\beta_k)\gamma_* \, G(y_k).\end{aligned} \tag{$\square$}$$

*The potential in (1) is non-increasing for the algorithm if the following conditions are satisfied.*

$$\gamma_{k+1} \le \frac{\beta_{k+1}}{2M\,\beta_k\,\gamma_k}\frac{(1-\beta_k)^2}{(1-\beta_{k+1})}, \tag{7}$$

$$1 - 2M\gamma_k^2 - M\beta_k^2\gamma_k^2 \ge 0, \tag{8}$$

$$\gamma_{k+1} \leq \frac{\beta_{k+1}}{2\,\beta_k(1-\beta_{k+1})} \left[ \frac{4M\gamma_k^2 + \beta_k^4 - 2\beta_k^3 + 3\beta_k^2 - 2}{(1-\beta_k)^2\big(M(\beta_k^2+2)\gamma_k^2 - 1\big)} \right] \gamma_k. \tag{9}$$

*With $\widetilde{c}_{k+1} \geq 0$ for $\theta = 2$, these give:*

$$V_k - V_{k+1} \geq \widetilde{c}_{k+1}L^2\|x_{k+1} - y_k\|^2.$$

*Proof.* The structure follows Lemma 1, with only the coupling changed.

$$x_{k+1} - x_k = \beta_k(x_0 - x_k) - \eta_k G(y_k), \tag{39}$$

$$x_{k+1} - x_k = \frac{\beta_k}{1-\beta_k}(x_0 - x_{k+1}) - \frac{\eta_k}{1-\beta_k}G(y_k), \tag{40}$$

and, using $\eta_k = (1 - \beta_k)\gamma_k$,

$$x_{k+1} - y_k = -\gamma_k\big[(1-\beta_k)G(y_k) - G(y_{k-1})\big]. \tag{41}$$

Monotonicity yields

$$\langle G(x_{k+1}), x_{k+1} - x_k \rangle \geq \langle G(x_k), x_{k+1} - x_k \rangle.$$

Substituting (39)–(40), rearranging, and multiplying by $b_k/\beta_k$ with $b_{k+1} = \frac{b_k}{1-\beta_k}$ gives

$$b_k\langle G(x_k), x_k - x_0 \rangle - b_{k+1}\langle G(x_{k+1}), x_{k+1} - x_0 \rangle \geq b_{k+1}\eta_k\langle G(x_{k+1}), G(y_k) \rangle + \frac{b_k\eta_k}{\beta_k}\langle G(x_{k+1}) - G(x_k), G(y_k) \rangle.$$

$$V_k - V_{k+1} \geq a_k\|G(x_k)\|^2 - a_{k+1}\|G(x_{k+1})\|^2 + c_kL^2\|x_k - y_{k-1}\|^2 - c_{k+1}L^2\|x_{k+1} - y_k\|^2$$
$$+ b_{k+1}\eta_k\langle G(x_{k+1}), G(y_k) \rangle + \frac{b_k\eta_k}{\beta_k}\langle G(x_{k+1}) - G(x_k), G(y_k) \rangle. \tag{42}$$

From Lipschitzness and (41), for any $\theta > 0$ we have

$$\|G(x_{k+1}) - G(y_k)\|^2 + \theta L^2\|x_{k+1} - y_k\|^2 \leq (1+\theta)L^2\|x_{k+1} - y_k\|^2$$
$$= (1+\theta)L^2\gamma_k^2\|(1-\beta_k)G(y_k) - G(y_{k-1})\|^2$$
$$\leq 2(1+\theta)L^2\gamma_k^2\|G(y_k) - G(y_{k-1})\|^2 + 2\beta_k^2(1+\theta)L^2\gamma_k^2\|G(y_k)\|^2. \tag{43}$$

Where we using $\|u + v\|^2 \leq 2\|u\|^2 + 2\|v\|^2$ to get the last inequality,

$$\|(1-\beta_k)G(y_k) - G(y_{k-1})\|^2 \leq 2\|G(y_k) - G(y_{k-1})\|^2 + 2\beta_k^2\|G(y_k)\|^2. \tag{44}$$

Using Lipschitzness between $x_k$ and $y_{k-1}$ we have:

$$\|G(y_k) - G(y_{k-1})\|^2 = \|G(y_k) - G(x_k) + G(x_k) - G(y_{k-1})\|^2$$
$$\leq 2\|G(x_k) - G(y_k)\|^2 + 2\|G(x_k) - G(y_{k-1})\|^2$$
$$\leq 2\|G(x_k)\|^2 - 4\langle G(x_k), G(y_k) \rangle + 2\|G(y_k)\|^2 + 2L^2\|x_k - y_{k-1}\|^2. \tag{45}$$

Expanding the left-hand side square of (43),

$$\|G(x_{k+1}) - G(y_k)\|^2 = \|G(x_{k+1})\|^2 - 2\langle G(x_{k+1}), G(y_k) \rangle + \|G(y_k)\|^2,$$

we obtain

$$\|G(x_{k+1})\|^2 - 2\langle G(x_{k+1}), G(y_k) \rangle + \|G(y_k)\|^2 + \theta L^2\|x_{k+1} - y_k\|^2$$
$$\leq 4L^2(1+\theta)\,\gamma_k^2\Big(\|G(x_k)\|^2 - 2\langle G(x_k), G(y_k) \rangle + \|G(y_k)\|^2\Big)$$
$$+ 4L^4(1+\theta)\,\gamma_k^2\,\|x_k - y_{k-1}\|^2 + 2L^2(1+\theta)\,\gamma_k^2\,\beta_k^2\|G(y_k)\|^2.$$

Setting $M := 2L^2(1 + \theta)$ and rearranging, we obtain the compact residual inequality

$$\|G(x_{k+1})\|^2 - 2\langle G(x_{k+1}), G(y_k)\rangle - 2M\gamma_k^2 \|G(x_k)\|^2 + 4M\gamma_k^2 \langle G(x_k), G(y_k)\rangle$$
$$+ \left(1 - 2M\gamma_k^2 - M\gamma_k^2\beta_k^2\right) \|G(y_k)\|^2 + \theta L^2 \|x_{k+1} - y_k\|^2 - 2L^2 M\gamma_k^2 \|x_k - y_{k-1}\|^2 \leq 0. \qquad (46)$$

Combine terms to isolate $G(x_{k+1}) - G(x_k)$:

$$\|G(x_{k+1})\|^2 - 2\left(1 - 2M\gamma_k^2\right) \langle G(x_{k+1}), G(y_k)\rangle - 2M\gamma_k^2 \|G(x_k)\|^2$$
$$- 4M\gamma_k^2 \langle G(x_{k+1}) - G(x_k), G(y_k)\rangle + \left(1 - 2M\gamma_k^2 - M\gamma_k^2\beta_k^2\right) \|G(y_k)\|^2$$
$$+ \theta L^2 \|x_{k+1} - y_k\|^2 - 2L^2 M\gamma_k^2 \|x_k - y_{k-1}\|^2 \leq 0.$$

Multiply the inequality obtained above by $\dfrac{a_k}{2M\gamma_k^2}$ and add it to Eq. (42). We get

$$V_k - V_{k+1} \geq \underbrace{\left(\frac{a_k}{2M\gamma_k^2} - a_{k+1}\right)}_{S_k^{11}} \|G(x_{k+1})\|^2$$

$$- 2 \underbrace{\left(\frac{a_k(1 - 2M\gamma_k^2)}{2M\gamma_k^2} - \frac{b_{k+1}\eta_k}{2}\right)}_{S_k^{12}} \langle G(x_{k+1}), G(y_k)\rangle$$

$$+ \underbrace{\left(\frac{a_k}{2M\gamma_k^2}\left[1 - 2M\gamma_k^2 - M\gamma_k^2\beta_k^2\right]\right)}_{S_k^{22}} \|G(y_k)\|^2$$

$$+ \underbrace{\left(-2a_k + \frac{b_k\eta_k}{\beta_k}\right)}_{S_k^{23}} \langle G(x_{k+1}) - G(x_k), G(y_k)\rangle$$

$$+ L^2 \underbrace{\left(c_k - a_k\right)}_{\tilde{c}_k} \|x_k - y_{k-1}\|^2$$

$$+ L^2 \underbrace{\left(\frac{a_k\theta}{2M\gamma_k^2} - c_{k+1}\right)}_{\tilde{c}_{k+1}} \|x_{k+1} - y_k\|^2. \qquad (47)$$

With the choice of $a_k = \dfrac{b_k\eta_k}{2\beta_k}$, and $c_k = a_k$, (and $b_{k+1} = \dfrac{b_k}{1 - \beta_k}$) we have $S_k^{23} = 0$, and $\tilde{c}_k = 0$.

From (47), to guarantee $V_k - V_{k+1} \geq 0$ it suffices to enforce $\tilde{c}_k \geq 0$, $\tilde{c}_{k+1} \geq 0$ and

$$S_k = \begin{pmatrix} S_k^{11} & S_k^{12} \\ S_k^{12} & S_k^{22} \end{pmatrix} \succeq 0, \quad \text{i.e.,} \quad S_k^{11} \geq 0, \ S_k^{22} \geq 0, \ S_k^{11} S_k^{22} \geq (S_k^{12})^2.$$

With $a_k = \frac{b_k\eta_k}{2\beta_k}$, $b_{k+1} = \frac{b_k}{1-\beta_k}$, and $\eta_k = (1 - \beta_k)\gamma_k$, the entries were

$$S_k^{11} = \frac{a_k}{2M\gamma_k^2} - a_{k+1} = \frac{b_k}{4M\beta_k} \frac{(1 - \beta_k)^2}{\eta_k} - \frac{b_k}{2(1 - \beta_k)} \frac{\eta_{k+1}}{\beta_{k+1}},$$

$$S_k^{12} = \frac{a_k(1 - 2M\gamma_k^2)}{2M\gamma_k^2} - \frac{b_{k+1}\eta_k}{2} = \frac{b_k}{4M\beta_k}\left(\frac{(1 - \beta_k)^2}{\eta_k} - 2M\eta_k\right) - \frac{b_k}{2(1 - \beta_k)}\eta_k,$$

$$S_k^{22} = \frac{a_k}{2M\gamma_k^2}\left[1 - 2M\gamma_k^2 - M\gamma_k^2\beta_k^2\right] = \frac{b_k}{4M\beta_k \eta_k}\left[(1 - \beta_k)^2 - M\eta_k^2(2 + \beta_k^2)\right].$$

$$S_k^{11} \geq 0 \iff \frac{(1-\beta_k)^2}{4M\beta_k\,\eta_k} \geq \frac{\eta_{k+1}}{2\beta_{k+1}(1-\beta_k)} \iff \eta_{k+1} \leq \frac{\beta_{k+1}}{2M\,\beta_k\,\eta_k}\,(1-\beta_k)^3. \tag{48}$$

In the $\gamma$-variables (using $\eta_k = (1-\beta_k)\gamma_k$ and $\eta_{k+1} = (1-\beta_{k+1})\gamma_{k+1}$), this is equivalent to

$$\gamma_{k+1} \leq \frac{\beta_{k+1}}{2M\,\beta_k\,\gamma_k}\,\frac{(1-\beta_k)^2}{(1-\beta_{k+1})}. \tag{49}$$

$$S_k^{22} \geq 0 \iff (1-\beta_k)^2 - M\eta_k^2\,(2+\beta_k^2) \geq 0 \iff \eta_k \leq \frac{1-\beta_k}{\sqrt{M\,(2+\beta_k^2)}}. \tag{50}$$

In $\gamma$-variables, this condition becomes

$$1 - 2M\gamma_k^2 - M\beta_k^2\gamma_k^2 \geq 0. \tag{51}$$

Assuming $S_k^{22} \geq 0$ and after rearrangement, we get the bound

$$S_k^{11}S_k^{22} \geq (S_k^{12})^2 \iff$$

$$\eta_{k+1} \leq \frac{\beta_{k+1}}{2M\,\beta_k\,(1-\beta_k)}\left[\frac{(1-\beta_k)^2}{\eta_k} - \frac{\left(\frac{(1-\beta_k)^2}{\eta_k} - 2M\,\eta_k - \frac{2M\beta_k}{1-\beta_k}\,\eta_k\right)^2}{\frac{(1-\beta_k)^2}{\eta_k} - M\,(2+\beta_k^2)\,\eta_k}\right]. \tag{52}$$

This simplifies to

$$\eta_{k+1} \leq \frac{\beta_{k+1}}{2\beta_k}\,\frac{\eta_k}{1-\beta_k}\,\frac{4M\eta_k^2 - (1-\beta_k)^3\left(\beta_k^3 - \beta_k^2 + 2\beta_k + 2\right)}{(1-\beta_k)^2(M(2+\beta_k^2)\,\eta_k^2 - (1-\beta_k)^2)}. \tag{53}$$

In the $\gamma$-variables, $\gamma_k := \eta_k/(1-\beta_k)$, (53) becomes

$$\gamma_{k+1} \leq \frac{\beta_{k+1}}{2\,\beta_k(1-\beta_{k+1})}\,\frac{4M\gamma_k^2 + \beta_k^4 - 2\beta_k^3 + 3\beta_k^2 - 2}{(1-\beta_k)^2(M(\beta_k^2+2)\gamma_k^2 - 1)}\,\gamma_k. \tag{54}$$

Conditions (49), (51), and (54), together with $\widetilde{c}_{k+1} = \frac{a_k\theta}{2M\gamma_k^2} - c_{k+1} \geq 0$, complete the proof of the one-step Lyapunov decrease.

Next, we show that conditions of Eq. (7), Eq. (8), Eq. (9) are satisfied, with constant step-size $\gamma_k = \gamma_* \in \left(0, \frac{2}{3\sqrt{10}\,L}\right)$, and the choice of $\beta_k = \frac{2}{k+6}$.

Let $\gamma_k \equiv \gamma \in \left(0, \frac{1}{\sqrt{3M}}\right)$ and $\beta_k = \frac{2}{k+6}$.

**Condition (8).** We need
$$1 - 2M\gamma^2 - M\beta_k^2\gamma^2 = 1 - M\gamma^2\,(2+\beta_k^2) \geq 0.$$

For $\beta_k \in [0,1]$, the right-hand side is minimized at $\beta_k = 1$, hence it suffices to require

$$M\gamma^2 \leq \frac{1}{3}.$$

Thus (8) holds for all $k$ whenever $M\gamma^2 \leq \frac{1}{3}$.

**Condition** (7). With $\gamma_{k+1} = \gamma_k = \gamma$, the condition (7) reads

$$(1 - \beta_{k+1})\gamma \ \leq \ \frac{\beta_{k+1}}{2M\,\beta_k\,\gamma}\,(1 - \beta_k)^2 \quad \Longleftrightarrow \quad 2M\gamma^2 \ \leq \ \frac{\beta_{k+1}(1 - \beta_k)^2}{\beta_k(1 - \beta_{k+1})}.$$

For the schedule $\beta_k = \frac{2}{k+6}$ we have $1 - \beta_k = \frac{k+4}{k+6}$, $\beta_{k+1} = \frac{2}{k+7}$, and $1 - \beta_{k+1} = \frac{k+5}{k+7}$. Hence

$$\frac{\beta_{k+1}(1 - \beta_k)^2}{\beta_k(1 - \beta_{k+1})} = \frac{\dfrac{2}{k+7}\left(\dfrac{k+4}{k+6}\right)^2}{\dfrac{2}{k+6}\left(\dfrac{k+5}{k+7}\right)} = \frac{(k+4)^2}{(k+5)(k+6)} \ < \ 1,$$

which is increasing in $k$, with minimum $\frac{8}{15}$ at $k = 0$ and limit $1$ as $k \to \infty$. Therefore

$$2M\gamma^2 \leq \frac{8}{15}$$

suffices for (7) to hold for all $k$. This is the binding requirement on the step-size.

**Condition** (9). With $\gamma_{k+1} = \gamma_k = \gamma$ the inequality (9) becomes

$$1 \ \leq \ \frac{\beta_{k+1}}{2\,\beta_k(1 - \beta_{k+1})} \frac{4M\gamma^2 + \beta_k^4 - 2\beta_k^3 + 3\beta_k^2 - 2}{(1 - \beta_k)^2(M(\beta_k^2 + 2)\gamma^2 - 1)}.$$

Let $t := M\gamma^2 \in (0, \frac{1}{2})$, $\beta = \beta_k$, and $\beta^+ = \beta_{k+1}$. Write

$$P_k := \frac{\beta^+}{2\beta(1 - \beta^+)}, \qquad Q_k(t) := \frac{4t + \overbrace{\left(\beta^4 - 2\beta^3 + 3\beta^2 - 2\right)}^{p(\beta)=:}}{(1 - \beta)^2\,(t(\beta^2 + 2) - 1)}.$$

For the schedule $\beta_k = \frac{2}{k+6}$ we have

$$P_k \ = \ \frac{\dfrac{2}{k+7}}{2 \cdot \dfrac{2}{k+6}\left(1 - \dfrac{2}{k+7}\right)} = \frac{k+6}{2(k+5)}.$$

Thus $P_k \in (\frac{1}{2}, \frac{3}{5}]$ and $P_k \downarrow \frac{1}{2}$ as $k \to \infty$.

On $[0, 1]$, the polynomial $p(\beta) = \beta^4 - 2\beta^3 + 3\beta^2 - 2$ is increasing and nonpositive, so for $\beta \in (0, \frac{1}{3}]$ we have $p(\beta) \leq p(1/3) < 0$. Since $t \leq \frac{1}{3}$, the numerator $4t + p(\beta)$ is strictly negative. The denominator $(1 - \beta)^2(t(\beta^2 + 2) - 1)$ is also strictly negative, so $Q_k(t) > 0$.

A direct calculation shows $\partial_\beta Q(\beta, t) \geq 0$ for $\beta \in (0, \frac{1}{3}]$, $t \in (0, \frac{1}{3}]$. Hence $Q_k(t)$ is increasing in $\beta$. Its minimum occurs as $\beta \downarrow 0$:

$$\lim_{\beta \downarrow 0} Q(\beta, t) \ = \ \frac{4t - 2}{2t - 1} \ = \ 2.$$

Therefore, for all admissible $(\beta, t)$,

$$Q_k(t) \ \geq \ 2.$$

Since $P_k > \frac{1}{2}$ and $Q_k(t) \geq 2$, we obtain

$$P_k\,Q_k(t) \ \geq \ \tfrac{1}{2} \cdot 2 = 1,$$

with strict inequality for every finite $k$.

Thus, condition (9) holds for all $k$ whenever $t = M\gamma^2 \leq \frac{1}{3}$.

**Positivity of the Lyapunov decrease.** We had

$$V_k - V_{k+1} \geq L^2 \left( \frac{a_k \theta}{2M\gamma_k^2} - a_{k+1} \right) \|x_{k+1} - y_k\|^2,$$

Recall $a_k = \frac{b_k \eta_k}{2\beta_k}$, $b_{k+1} = \frac{b_k}{1-\beta_k}$, and $\eta_k = (1 - \beta_k)\gamma$ with constant $\gamma$. Then

$$\frac{a_{k+1}}{a_k} = \frac{b_{k+1}}{b_k} \cdot \frac{\eta_{k+1}}{\eta_k} \cdot \frac{\beta_k}{\beta_{k+1}} = \frac{\beta_k(1 - \beta_{k+1})}{\beta_{k+1}(1 - \beta_k)^2}.$$

Hence the residual coefficient is nonnegative iff

$$\frac{\theta}{2M\gamma^2} \geq \frac{a_{k+1}}{a_k} = \frac{\beta_k(1 - \beta_{k+1})}{\beta_{k+1}(1 - \beta_k)^2}.$$

For the schedule $\beta_k = \frac{2}{k+6}$ we have

$$\frac{a_{k+1}}{a_k} = \frac{(k+5)(k+6)}{(k+4)^2}, \quad \text{whose supremum is } \frac{15}{8} \text{ at } k = 0.$$

Therefore it suffices to require

$$2M\gamma^2 \leq \frac{8}{15}\theta.$$

In particular by choosing $\theta = 2$, this condition can be satisfied, giving $M = 2L^2(1 + \theta) = 6L^2$. Collecting the step-size requirements with $\theta = 2$, $M = 6L^2$,

$$\underbrace{2M\gamma^2 \leq \frac{8}{15}}_{(7)}, \qquad \underbrace{M\gamma^2 \leq \frac{1}{3}}_{(8),(9)}, \qquad \underbrace{2M\gamma^2 \leq \frac{8}{15}\theta = \frac{16}{15}}_{\text{residual}},$$

the binding constraint is (7), i.e. $\gamma^2 \leq \frac{8}{15 \cdot 2M} = \frac{2}{45L^2}$. This yields the step-size range $\gamma_* \in \left( 0, \frac{2}{3\sqrt{10}\,L} \right)$.

$\square$

## A.4. Proof of Theorem 2

**Theorem 2.** *Suppose $G$ is monotone and $L$-Lipschitz, and let $x^\star$ satisfy $G(x^\star) = 0$. Consider the update of ($\square$). If the step-size $\gamma$ and anchoring coefficient $\beta_k$ satisfy the conditions of Lemma 2, then the Lyapunov function of Eq. (1) is decreasing. This implies the bound*

$$\|G(x_k)\|^2 \leq \frac{16/\gamma_*^2 + 32L^2}{(k+4)^2} \|x_0 - x^\star\|^2. \tag{10}$$

*Moreover, the constant $16/\gamma_*^2 + 32L^2$ is decreasing in $\gamma_*$, so it is smallest at the largest admissible step; as $\gamma_* \to \frac{2}{3\sqrt{10}\,L}$ it gives*

$$\|G(x_k)\|^2 \leq \frac{392L^2}{(k+4)^2} \|x_0 - x^\star\|^2. \tag{11}$$

*Proof.* Similar to the proof of theorem 1 define

$$H_k := a_k \|G(x_k)\|^2 + b_k \langle G(x_k), x_k - x_0 \rangle.$$

By monotonicity and Young's inequality $\langle a, b \rangle \leq \frac{\alpha}{2}\|a\|^2 + \frac{1}{2\alpha}\|b\|^2$ with $\alpha = a_k$, we obtain

$$H_k = a_k \|G(x_k)\|^2 + b_k \langle G(x_k) - G(x^\star), x_k - x^\star \rangle + b_k \langle G(x_k), x^\star - x_0 \rangle$$

$$\geq a_k \|G(x_k)\|^2 - \frac{a_k}{2}\|G(x_k)\|^2 - \frac{b_k^2}{2a_k}\|x_0 - x^\star\|^2$$

$$= \frac{a_k}{2}\|G(x_k)\|^2 - \frac{b_k^2}{2a_k}\|x_0 - x^\star\|^2. \tag{55}$$

From Eq. (55), it follows that

$$\frac{a_k}{2}\|G(x_k)\|^2 \leq H_k + \frac{b_k^2}{2a_k}\|x_0 - x^\star\|^2,$$

which implies

$$\frac{a_k}{2}\|G(x_k)\|^2 + c_k L^2\|x_k - y_{k-1}\|^2 \leq V_k + \frac{b_k^2}{2a_k}\|x_0 - x^\star\|^2. \tag{56}$$

Since $V_k$ is decreasing by Lemma 2, we have $V_k \leq V_0$. Substituting the explicit expressions for $a_k, b_k, c_k$ and using $y_{-1} = x_0$, inequality Eq. (56) yields

$$\frac{b_0 \gamma_*}{160}(k+5)(k+4)^2 \|G(x_k)\|^2 \leq V_0 + \frac{b_k^2}{2a_k}\|x_0 - x^\star\|^2$$

$$= V_0 + \frac{b_0(k+5)}{10\,\gamma_*}\|x_0 - x^\star\|^2.$$

Bounding the initial potential by $V_0 = a_0\|G(x_0)\|^2 = b_0\gamma_*\|G(x_0)\|^2 \leq b_0\gamma_* L^2\|x_0 - x^\star\|^2$ and multiplying both sides by $\frac{160}{b_0\gamma_*(k+5)(k+4)^2}$ gives

$$\|G(x_k)\|^2 \;\leq\; \frac{160L^2}{(k+4)^2(k+5)}\|x_0 - x^\star\|^2 + \frac{16}{\gamma_*^2(k+4)^2}\|x_0 - x^\star\|^2.$$

Using $\frac{160L^2}{(k+4)^2(k+5)} \leq \frac{32L^2}{(k+4)^2}$, which holds for every $k \geq 0$ (with equality at $k=0$), we obtain

$$\|G(x_k)\|^2 \;\leq\; \frac{16/\gamma_*^2 + 32L^2}{(k+4)^2}\|x_0 - x^\star\|^2,$$

which establishes Eq. (10). Finally, taking $\gamma_* \to \frac{2}{3\sqrt{10}\,L}$ gives $16/\gamma_*^2 \to 360L^2$, hence

$$\|G(x_k)\|^2 \;\leq\; \frac{392L^2}{(k+4)^2}\|x_0 - x^\star\|^2.$$

$\square$

### A.5. Analysis of the Bilinear Game

**Example.** Let $f(x,y) = Lxy$, with operator $G(x,y) = (Ly, -Lx)$ and solution $x^\star = (0,0)$. Fix $a, b > 0$ with $b > a$, constant step size $\eta_\star = \frac{1}{\theta L}$ ($\theta > 0$), and anchoring coefficient $\beta_k = \frac{a}{k+b}$. For any $(p,q) \in \mathbb{R}^2$, initialize ($\triangle$) at $x_0 = \frac{a\theta}{b-a}(-q, p)$. Then

$$\|G(x_k)\|^2 = \frac{(a\theta L)^2}{(k+b-a)^2}\|x_0 - x^\star\|^2 \left(1 + o(1)\right). \tag{57}$$

*Proof.* The bilinear operator satisfies the two elementary identities

$$\|G(v)\| = L\,\|v\|, \qquad G^2 = -L^2 I, \qquad v \in \mathbb{R}^2, \tag{P}$$

both immediate from $G(x,y) = (Ly, -Lx)$.

Multiplying the two lines of ($\triangle$) by $(k+b)$ and using $(k+b)\beta_k = a$, $(k+b)(1-\beta_k) = k+b-a$, then setting the scaled variables $\tilde{x}_k := (k+b-a)x_k$ and $\tilde{y}_k := (k+b)y_k$, linearity of $G$ gives the exact system

$$\tilde{y}_k = \tilde{x}_k + ax_0 - \eta_\star \tfrac{k+b-a}{k+b-1} G(\tilde{y}_{k-1}), \tag{58}$$

$$\tfrac{k+b}{k+b-a+1}\tilde{x}_{k+1} = \tilde{x}_k + ax_0 - \eta_\star G(\tilde{y}_k). \tag{59}$$

Both prefactors tend to 1 (and equal 1 for all $k$ iff $a = 1$), so the limiting system is time-invariant with fixed point $\tilde{x}_k = \tilde{y}_k \equiv \tilde{x}$ given by $\eta_\star G(\tilde{x}) = a x_0$.

Hence $G(\tilde{x}) = a\theta L\, x_0$, and taking norms via $\|G(\tilde{x})\| = L\|\tilde{x}\|$ from (P),

$$\|\tilde{x}\| = a\theta \, \|x_0\|. \tag{60}$$

It remains to show $\tilde{x}_k \to \tilde{x}$. With $d_k := \tilde{x}_k - \tilde{x}$ and $A := \eta_\star G$ (so $A^2 = -g^2 I$, $g := \eta_\star L = 1/\theta$, by (P)), the deviation obeys the deviation obeys $d_{k+1} = (I - 2A)d_k + A\, d_{k-1}$ (subtracting the fixed point from the limiting system gives $e_k = d_k - A e_{k-1}$ and $d_{k+1} = d_k - A e_k$ with $e_k := \tilde{y}_k - \tilde{x}$; eliminate $e_k$ using $A^{-1} = -g^{-2}A$).. Since $\{I, A\}$ commute and $A^2 = -g^2 I$, the characteristic roots have $|\lambda_\pm|^2 = r_\pm := \frac{1 \pm \sqrt{1-4g^2}}{2}$. Any admissible step $\eta_\star \leq \frac{1}{2\sqrt{3}L}$ gives $4g^2 \leq \frac{1}{3} < 1$, hence $r_\pm \in (0,1)$ and $|\lambda_\pm| < 1$: the homogeneous map is a contraction. As the prefactors in (58)–(59) differ from 1 by $O(1/k) \to 0$, it stays a contraction for large $k$, so $d_k \to 0$.

Finally, from $x_k = \tilde{x}_k/(k+b-a)$, $\tilde{x}_k \to \tilde{x}$, $\|G(x_k)\| = L\|x_k\|$, and (60),

$$\|G(x_k)\|^2 = \frac{L^2\|\tilde{x}_k\|^2}{(k+b-a)^2} \longrightarrow \frac{(a\theta L)^2}{(k+b-a)^2}\,\|x_0\|^2,$$

which is (57) since $x^\star = 0$. $\qquad\square$

**Comparison with the general guarantee.** With $a = 2$, $b = 6$ and the largest admissible step $\eta_\star = \frac{1}{2\sqrt{3}L}$ ($\theta = 2\sqrt{3}$), we have $k+b-a = k+4$ and $(a\theta L)^2 = 48L^2$, so $\|G(x_k)\|^2 = \frac{48L^2}{(k+4)^2}\|x_0 - x^\star\|^2(1 + o(1))$. The universal bound (6), with $1/\eta_\star^2 = \theta^2 L^2$, equals $\frac{264L^2}{(k+6)^2}$. Both decay as $L^2/k^2$, and the ratio of leading constants is independent of $k$:

$$\left.\frac{16\theta^2 + 72}{(a\theta)^2}\right|_{a=2} = 4 + \frac{18}{\theta^2} \xrightarrow{\theta=2\sqrt{3}} \frac{11}{2} = 5.5.$$

So on this hard instance the universal analysis is tight up to a constant factor at most 5.5.

# B. Proof of Section 5

## B.1. Setup

Throughout this section, $G$ is monotone and $L$-Lipschitz with $G(x^\star) = 0$, and $\widehat{G}(x, \xi)$ satisfies Assumption 3, where, as argued after Assumption 3, we may take $\kappa \geq 1$ without loss of generality. We write $d_0 := \|x_0 - x^\star\|$ and let $\mathcal{F}_k := \sigma(x_0, \xi_0, \ldots, \xi_{k-1})$ denote the natural filtration of the algorithm, so that $x_k, y_k \in \mathcal{F}_k$. We consider the stochastic updates (14) with the schedules

$$\beta_k = \frac{1}{k+2}, \qquad \eta_k = \frac{1}{L\sqrt{\kappa}\,(k+2)^{3/4}}, \tag{61}$$

for which

$$L^2\eta_k^2 = \frac{1}{\kappa(k+2)^{3/2}} \leq \frac{1}{(k+2)^{3/2}}, \qquad \sum_{k\geq 0} L^2\eta_k^2 \leq \frac{1}{\kappa}\sum_{m\geq 2} m^{-3/2} \leq \frac{2}{\kappa} \leq 2, \tag{62}$$

together with the deterministic reference trajectory $(\bar{x}_k, \bar{y}_k)$ defined in (15). Since $\|G(x_N)\|^2 \leq 2\|G(\bar{x}_N)\|^2 + 2L^2\|x_N - \bar{x}_N\|^2$ by $L$-Lipschitzness, Theorem 4 follows from Lemma 3 (proved in Appx. B.2) and Lemma 4 (proved in Appx. B.3), both stated in Section 5.

## B.2. Proof of Lemma 3 (deterministic reference bound)

**Lemma 3.** *Let $G$ be monotone and $L$-Lipschitz with $G(x^\star) = 0$, and let $(\bar{x}_k, \bar{y}_k)$ be given by (15) with $\beta_k = \frac{1}{k+2}$, $\eta_k = \frac{1}{L\sqrt{\kappa}(k+2)^{3/4}}$, and $\kappa \geq 1$. Then for all $N \geq 1$ and all $k \geq 0$,*

$$\|G(\bar{x}_N)\|^2 \leq \frac{33\,L^2\kappa\,\|x_0 - x^\star\|^2}{\sqrt{N+1}}, \tag{16}$$

$$\|G(\bar{y}_k)\|^2 \leq \frac{94\,L^2\kappa\,\|x_0 - x^\star\|^2}{\sqrt{k+1}}. \tag{17}$$

*Proof.* **Step 1: Lyapunov potential and one-step decrease.** Consider the two-term anchored Lyapunov potential evaluated along the reference trajectory, namely the analogue of the deterministic potential (1) in which the extrapolation term $c_k L^2 \|x_k - y_{k-1}\|^2$ is dropped, since the simplified variant sets $\gamma_k = 0$ and does not reuse past gradients:

$$H_k := a_k \|G(\bar{x}_k)\|^2 + b_k \langle G(\bar{x}_k), \bar{x}_k - x_0 \rangle, \qquad a_{k+1} = \frac{b_k \eta_k}{2\beta_k(1 - \beta_k)}, \qquad b_{k+1} = \frac{b_k}{1 - \beta_k}. \tag{63}$$

The first term is the residual we control; the second is an anchoring cross-term coupling the gradient with the displacement from the reference point $x_0$. For the schedules (61), normalizing $b_0 = 1$,

$$b_k = k + 1, \qquad a_k = \frac{(k+1)^{5/4}}{2L\sqrt{\kappa}} \quad (k \geq 1), \qquad \text{and} \qquad b_1 \eta_0 = a_1. \tag{64}$$

From (15) we have the identities

$$\bar{x}_{k+1} - \bar{x}_k = \beta_k(x_0 - \bar{x}_k) - \eta_k G(\bar{y}_k), \qquad \bar{x}_{k+1} - \bar{x}_k = \tfrac{\beta_k}{1-\beta_k}(x_0 - \bar{x}_{k+1}) - \tfrac{\eta_k}{1-\beta_k}G(\bar{y}_k).$$

Substituting both into the monotonicity inequality $\langle G(\bar{x}_{k+1}) - G(\bar{x}_k), \bar{x}_{k+1} - \bar{x}_k \rangle \geq 0$, rearranging, and multiplying by $b_k/\beta_k$ (with $b_{k+1} = \frac{b_k}{1-\beta_k}$) yields

$$b_k \langle G(\bar{x}_k), \bar{x}_k - x_0 \rangle - b_{k+1} \langle G(\bar{x}_{k+1}), \bar{x}_{k+1} - x_0 \rangle \geq \tfrac{b_k \eta_k}{\beta_k(1-\beta_k)}\langle G(\bar{x}_{k+1}), G(\bar{y}_k) \rangle - \tfrac{b_k \eta_k}{\beta_k}\langle G(\bar{x}_k), G(\bar{y}_k) \rangle. \tag{65}$$

By $L$-Lipschitzness and $\bar{x}_{k+1} - \bar{y}_k = -\eta_k G(\bar{y}_k)$,

$$\|G(\bar{x}_{k+1})\|^2 - 2\langle G(\bar{x}_{k+1}), G(\bar{y}_k) \rangle + (1 - L^2\eta_k^2)\|G(\bar{y}_k)\|^2 \leq 0. \tag{66}$$

Adding the $a_k$-terms to (65) and then adding $a_{k+1} \times$(66), the $\|G(\bar{x}_{k+1})\|^2$ terms and the $\langle G(\bar{x}_{k+1}), G(\bar{y}_k) \rangle$ terms cancel (using $\frac{b_k \eta_k}{\beta_k(1-\beta_k)} = 2a_{k+1}$), leaving the one-step inequality

$$H_k - H_{k+1} \geq a_k \|G(\bar{x}_k)\|^2 - 2a_{k+1}(1 - \beta_k)\langle G(\bar{x}_k), G(\bar{y}_k) \rangle + a_{k+1}(1 - L^2\eta_k^2)\|G(\bar{y}_k)\|^2. \tag{67}$$

The right-hand side of (67) is a quadratic form in $(\|G(\bar{x}_k)\|, \|G(\bar{y}_k)\|)$ and is nonnegative provided

$$a_k a_{k+1}(1 - L^2\eta_k^2) \geq a_{k+1}^2(1 - \beta_k)^2, \qquad \text{i.e.} \qquad 1 - L^2\eta_k^2 \geq \left(\frac{k+1}{k+2}\right)^{3/4}, \tag{68}$$

using $a_{k+1}/a_k = \big((k+2)/(k+1)\big)^{5/4}$ from (64). By the tangent bound $\big(1 - \frac{1}{k+2}\big)^{3/4} \leq 1 - \frac{3}{4(k+2)}$ (concavity of $t \mapsto t^{3/4}$) and $L^2\eta_k^2 \leq (k+2)^{-3/2}$ from (62), condition (68) holds as soon as $(k+2)^{-3/2} \leq \frac{3}{4(k+2)}$, i.e. $(k+2)^{1/2} \geq \frac{4}{3}$, which is true for every $k \geq 0$. Hence $H_{k+1} \leq H_k$ for all $k \geq 1$, so

$$H_N \leq H_1 \qquad (N \geq 1). \tag{69}$$

(Starting the telescoping at $k = 1$ sidesteps the $a_0 = 0$ boundary step entirely.)

**Step 2: lower bound on $H_N$.** Since $G(x^\star) = 0$, monotonicity gives $\langle G(\bar{x}_N), \bar{x}_N - x^\star \rangle \geq 0$, so writing $\bar{x}_N - x_0 = (\bar{x}_N - x^\star) + (x^\star - x_0)$ and applying Young's inequality,

$$H_N \geq a_N \|G(\bar{x}_N)\|^2 - b_N \|G(\bar{x}_N)\| \, d_0 \geq \frac{a_N}{2}\|G(\bar{x}_N)\|^2 - \frac{b_N^2}{2a_N}d_0^2. \tag{70}$$

Combining (69) and (70),

$$\|G(\bar{x}_N)\|^2 \leq \frac{2H_1}{a_N} + \left(\frac{b_N}{a_N}\right)^2 d_0^2, \qquad \left(\frac{b_N}{a_N}\right)^2 = \frac{4L^2\kappa}{\sqrt{N+1}}. \tag{71}$$

**Step 3: the term $H_1/a_N$ is of lower order.** Since $\beta_0 = \frac{1}{2}$ and $\bar{x}_0 = x_0$, we have $\bar{y}_0 = x_0$ and $\bar{x}_1 = x_0 - \eta_0 G(x_0)$. As $L\eta_0 \leq 1$ by (62), Lipschitzness gives $\|G(\bar{x}_1)\| \leq (1 + L\eta_0)\|G(x_0)\| \leq 2Ld_0$, and

$$\langle G(\bar{x}_1), \bar{x}_1 - x_0 \rangle = -\eta_0 \langle G(\bar{x}_1), G(x_0) \rangle \leq \eta_0 \|G(\bar{x}_1)\| \, \|G(x_0)\| \leq 2\eta_0 L^2 d_0^2.$$

With $b_1\eta_0 = a_1$ this yields $H_1 \leq 4a_1 L^2 d_0^2 + 2b_1\eta_0 L^2 d_0^2 = 6a_1 L^2 d_0^2$, and since $a_1/a_N = \left(2/(N+1)\right)^{5/4}$,

$$\frac{2H_1}{a_N} \leq \frac{12 \cdot 2^{5/4} L^2 d_0^2}{(N+1)^{5/4}}, \tag{72}$$

Plugging (72) into (71) with $\kappa \geq 1$ proves (16).

**Step 4: boundedness of the reference iterates.** Since $\bar{x}_{k+1} = \bar{y}_k - \eta_k G(\bar{y}_k)$, monotonicity and Lipschitzness (with $G(x^\star) = 0$) give

$$\|\bar{x}_{k+1} - x^\star\|^2 = \|\bar{y}_k - x^\star\|^2 - 2\eta_k\langle G(\bar{y}_k), \bar{y}_k - x^\star\rangle + \eta_k^2\|G(\bar{y}_k)\|^2 \leq \left(1 + L^2\eta_k^2\right)\|\bar{y}_k - x^\star\|^2.$$

Together with $\|\bar{y}_k - x^\star\| \leq \beta_k d_0 + (1 - \beta_k)\|\bar{x}_k - x^\star\|$ and (62), an immediate induction gives, for all $k$,

$$\|\bar{x}_k - x^\star\| \leq \prod_{j\geq 0}\left(1 + L^2\eta_j^2\right)^{1/2} d_0 \leq e^{\frac{1}{2}\sum_j L^2\eta_j^2} d_0 \leq e\, d_0, \qquad \|\bar{y}_k - x^\star\| \leq e\, d_0, \tag{73}$$

and consequently

$$\|\bar{y}_k - \bar{x}_k\| = \beta_k\|x_0 - \bar{x}_k\| \leq (1 + e)\beta_k d_0. \tag{74}$$

**Step 5: residual bound at $\bar{y}_k$.** For $k \geq 1$, Lipschitzness, the bound (16) applied with $N = k$, and (74) give

$$\|G(\bar{y}_k)\|^2 \leq 2\|G(\bar{x}_k)\|^2 + 2L^2\|\bar{y}_k - \bar{x}_k\|^2 \leq \frac{66L^2\kappa d_0^2}{\sqrt{k+1}} + \frac{2(1+e)^2 L^2 d_0^2}{(k+2)^2} \leq \frac{\left(66 + 2(1+e)^2\right)L^2\kappa d_0^2}{\sqrt{k+1}},$$

since $(k+2)^2 \geq \sqrt{k+1}$ and $\kappa \geq 1$. For $k = 0$, $\bar{y}_0 = x_0$ and $\|G(x_0)\|^2 \leq L^2 d_0^2$. This proves (17) with $66 + 2(1+e)^2 \leq 94$. $\qquad\square$

### B.3. Proof of Lemma 4 (stochastic stability)

**Lemma 4.** *In the setting of Lemma 3, let $\widehat{G}$ satisfy Assumption 3, let $(x_k)$ be given by (14), and set $e_k := x_k - \bar{x}_k$. Then for all $N \geq 0$,*

$$\mathbb{E}\|e_N\|^2 \leq \frac{1}{\sqrt{N+1}}\left(\frac{4\sigma^2}{L^2\kappa} + 752\,\kappa\,\|x_0 - x^\star\|^2\right). \tag{18}$$

*Proof.* **Step 1: error recursion.** Let $n_k := \widehat{G}(y_k, \xi_k) - G(y_k)$, so $\mathbb{E}[n_k \mid \mathcal{F}_k] = 0$ and, by Assumption 3 and unbiasedness,

$$\mathbb{E}\left[\|n_k\|^2 \mid \mathcal{F}_k\right] = \mathbb{E}\left[\|\widehat{G}(y_k, \xi_k)\|^2 \mid \mathcal{F}_k\right] - \|G(y_k)\|^2 \leq \sigma^2 + \kappa\|G(y_k)\|^2. \tag{75}$$

Subtracting the reference update (15) from the stochastic update (14), and using $y_k - \bar{y}_k = (1 - \beta_k)e_k$,

$$e_{k+1} = (1 - \beta_k)e_k - \eta_k\left(G(y_k) - G(\bar{y}_k)\right) - \eta_k n_k. \tag{76}$$

The first two terms on the right-hand side are $\mathcal{F}_k$-measurable, so taking conditional expectations and using $\mathbb{E}[n_k \mid \mathcal{F}_k] = 0$,

$$\mathbb{E}\left[\|e_{k+1}\|^2 \mid \mathcal{F}_k\right] = \left\|(1 - \beta_k)e_k - \eta_k\left(G(y_k) - G(\bar{y}_k)\right)\right\|^2 + \eta_k^2\,\mathbb{E}\left[\|n_k\|^2 \mid \mathcal{F}_k\right]. \tag{77}$$

**Step 2: the drift term is contractive.** By monotonicity, $\langle G(y_k) - G(\bar{y}_k), y_k - \bar{y}_k\rangle \geq 0$, and since $y_k - \bar{y}_k = (1 - \beta_k)e_k$ with $\beta_k < 1$,

$$\langle G(y_k) - G(\bar{y}_k), e_k\rangle \geq 0. \tag{78}$$

By Lipschitzness, $\|G(y_k) - G(\bar{y}_k)\| \leq L(1 - \beta_k)\|e_k\|$. Expanding the square and using (78) to drop the cross term,

$$\left\|(1 - \beta_k)e_k - \eta_k\left(G(y_k) - G(\bar{y}_k)\right)\right\|^2 \leq (1 - \beta_k)^2\left(1 + L^2\eta_k^2\right)\|e_k\|^2. \tag{79}$$

Moreover, again by Lipschitzness,

$$\|G(y_k)\|^2 \leq 2\|G(\bar{y}_k)\|^2 + 2L^2(1 - \beta_k)^2\|e_k\|^2. \tag{80}$$

**Step 3: one-step inequality.** Combining (77)–(80) with (75),

$$\mathbb{E}\big[\|e_{k+1}\|^2 \mid \mathcal{F}_k\big] \leq q_k \|e_k\|^2 + \eta_k^2 \sigma^2 + 2\kappa \eta_k^2 \|G(\bar{y}_k)\|^2, \qquad q_k := (1 - \beta_k)^2 \Big(1 + (1 + 2\kappa) L^2 \eta_k^2\Big). \tag{81}$$

Since $\kappa \geq 1$ implies $(1 + 2\kappa)/\kappa \leq 3$, the schedule (61) gives $(1 + 2\kappa) L^2 \eta_k^2 \leq 3(k + 2)^{-3/2}$, hence, with $m := k + 2 \geq 2$,

$$q_k \leq \Big(1 - \frac{1}{m}\Big)^2 \Big(1 + \frac{3}{m^{3/2}}\Big) \leq 1 - \frac{3}{4m}. \tag{82}$$

The second inequality in (82) is elementary: it is equivalent to $g(m) := 3m^{-1/2}(1 - 1/m)^2 + 1/m \leq \frac{5}{4}$, which holds for $m \geq 9$ since then $3m^{-1/2} \leq 1$ and $1/m \leq \frac{1}{9}$, and is verified directly for $m \in \{2, \ldots, 8\}$ (its maximum there is $g(3) \leq 1.11$).

**Step 4: solving the recursion.** Let $r_k := \mathbb{E}\|e_k\|^2$, so $r_0 = 0$. Taking total expectations in (81), with $\eta_k^2 \sigma^2 = \sigma^2/(L^2 \kappa(k + 2)^{3/2})$ and, by (17) and $\sqrt{k + 1} \geq 1$,

$$2\kappa \eta_k^2 \|G(\bar{y}_k)\|^2 \leq \frac{2\kappa}{L^2 \kappa(k + 2)^{3/2}} \cdot \frac{94 L^2 \kappa d_0^2}{\sqrt{k + 1}} \leq \frac{188 \kappa d_0^2}{(k + 2)^{3/2}},$$

we obtain

$$r_{k+1} \leq \Big(1 - \frac{3}{4(k + 2)}\Big) r_k + \frac{S}{(k + 2)^{3/2}}, \qquad S := \frac{\sigma^2}{L^2 \kappa} + 188 \kappa d_0^2. \tag{83}$$

We claim that (83) with $r_0 = 0$ implies $r_k \leq 4S/\sqrt{k + 2}$ for all $k \geq 0$. The base case is immediate; if $r_k \leq 4S\, m^{-1/2}$ with $m = k + 2$, then

$$r_{k+1} \leq \Big(1 - \frac{3}{4m}\Big) \frac{4S}{\sqrt{m}} + \frac{S}{m^{3/2}} = \frac{4S}{\sqrt{m}} - \frac{2S}{m^{3/2}} \leq \frac{4S}{\sqrt{m + 1}},$$

where the last step uses

$$\frac{1}{\sqrt{m}} - \frac{1}{\sqrt{m + 1}} = \frac{1}{\sqrt{m}\,\sqrt{m + 1}\,(\sqrt{m} + \sqrt{m + 1})} \leq \frac{1}{2m^{3/2}}.$$

This proves $r_N \leq \frac{4S}{\sqrt{N+1}}$, which is exactly (18). $\qquad \square$

### B.4. Proof of Theorem 4

**Theorem 4.** *Let $G : \mathbb{R}^d \to \mathbb{R}^d$ be monotone and $L$-Lipschitz and $\widehat{G}(x, \xi)$ be a stochastic oracle following Assumption 3. Then for the updates described in (14) with $\beta_k = \frac{1}{k+2}$ and $\eta_k = \frac{1}{L\sqrt{\kappa}(k+2)^{3/4}}$, we have for all $N \geq 0$,*

$$\mathbb{E}\|G(x_N)\|^2 \leq \frac{1570 L^2 \kappa \|x_0 - x^\star\|^2}{\sqrt{N + 1}} + \frac{8\,\sigma^2}{\kappa \sqrt{N + 1}}.$$

*Proof of Theorem 4.* For $N = 0$ the bound holds trivially: $\|G(x_0)\|^2 \leq L^2 d_0^2 \leq 1570 L^2 \kappa d_0^2$ since $\kappa \geq 1$. Let $N \geq 1$. By $L$-Lipschitzness of $G$,

$$\|G(x_N)\|^2 \leq 2\|G(\bar{x}_N)\|^2 + 2L^2 \|x_N - \bar{x}_N\|^2.$$

Taking expectations and applying Lemma 3 and Lemma 4,

$$\mathbb{E}\,\|G(x_N)\|^2 \leq \frac{66 L^2 \kappa d_0^2}{\sqrt{N + 1}} + \frac{2L^2}{\sqrt{N + 1}} \Big(\frac{4\sigma^2}{L^2 \kappa} + 752 \kappa d_0^2\Big) = \frac{1570 L^2 \kappa d_0^2}{\sqrt{N + 1}} + \frac{8\sigma^2}{\kappa \sqrt{N + 1}}.$$

$\qquad \square$

### B.5. Proof of Non-stochastic Rate of $(\diamond)$

**Theorem 5.** *Assume $G$ is monotone and $L$-Lipschitz. Consider*

$$y_k = \beta_k x_0 + (1 - \beta_k)x_k, \qquad x_{k+1} = y_k - \eta_k\, G(y_k),$$

*with $(y_{-1} = x_0)$. Potential of (1) with the coefficients $a_{k+1} = \frac{b_k \eta_k}{2\beta_k(1-\beta_k)}$ and $b_{k+1} = \frac{b_k}{1-\beta_k}$, $c_k = a_k$, and the choice of parameters:*

$$\eta_k = \frac{c}{L}\sqrt{\frac{\beta_k}{2}}, \quad c \in (0, \frac{1}{\sqrt{2}}], \quad \beta_k = \frac{1}{k+2}.$$

*admits, with $\widetilde{c}_{k+1} \geq 0$ for $\theta = 1$, the one-step decrease $V_k - V_{k+1} \geq \widetilde{c}_{k+1}L^2\|x_{k+1} - y_k\|^2$, hence $V_{k+1} \leq V_k$ for all $k \geq 1$. Consequently, for all $k \geq 1$,*

$$\|G(x_k)\|^2 \;\leq\; \frac{C_{\mathrm{init}}\,L^2}{(k+1)^{3/2}}\,\|x_0 - x^\star\|^2 \;+\; \frac{C_\star\, L^2}{k+1}\|x_0 - x^\star\|^2.$$

*In particular, $\|G(x_k)\|^2 = \mathcal{O}((k+1)^{-1})$.*

*Proof.* From the update rules we obtain:

$$x_{k+1} - x_k = \beta_k(x_0 - x_k) - \eta_k G(y_k), \tag{84}$$

$$x_{k+1} - x_k = \frac{\beta_k}{1 - \beta_k}(x_0 - x_{k+1}) - \frac{\eta_k}{1 - \beta_k}G(y_k), \tag{85}$$

$$x_{k+1} - y_k = -\eta_k G(y_k). \tag{86}$$

By monotonicity of $G$:

$$\langle G(x_{k+1}) - G(x_k),\, x_{k+1} - x_k \rangle \geq 0$$

$$\langle G(x_{k+1}),\, x_{k+1} - x_k \rangle \geq \langle G(x_k),\, x_{k+1} - x_k \rangle$$

Using Eq. (84) and Eq. (85), we write:

$$\langle G(x_{k+1}),\, \frac{\beta_k}{1 - \beta_k}(x_0 - x_{k+1}) - \frac{\eta_k}{1 - \beta_k}G(y_k) \rangle \geq \langle G(x_k),\, \beta_k(x_0 - x_k) - \eta_k G(y_k) \rangle$$

Rearranging:

$$\frac{\beta_k}{1 - \beta_k}\langle G(x_{k+1}),\, x_0 - x_{k+1}\rangle \geq \beta_k\langle G(x_k),\, x_0 - x_k\rangle - \eta_k\langle G(x_k),\, G(y_k)\rangle + \frac{\eta_k}{1 - \beta_k}\langle G(x_{k+1}),\, G(y_k)\rangle$$

Multiplying this inequality by $\frac{b_k}{\beta_k}$ and taking $b_{k+1} = \frac{b_k}{1-\beta_k}$:

$$b_k\langle G(x_k), x_k - x_0\rangle - b_{k+1}\langle G(x_{k+1}), x_{k+1} - x_0\rangle \geq \frac{b_k\,\eta_k}{\beta_k(1 - \beta_k)}\langle G(x_{k+1}), G(y_k)\rangle - \frac{b_k\,\eta_k}{\beta_k}\langle G(x_k), G(y_k)\rangle \tag{87}$$

Recall:

$$V_k = a_k\|G(x_k)\|^2 + b_k\langle G(x_k),\, x_k - x_0\rangle + c_k L^2\|x_k - y_{k-1}\|^2.$$

Adding $a_k$ and $c_k$ terms to Eq. (87) gives

$$
\begin{aligned}
V_k - V_{k+1} \geq a_k \|G(x_k)\|^2 - a_{k+1}\|G(x_{k+1})\|^2 + \frac{b_k \eta_k}{\beta_k(1-\beta_k)}\langle G(x_{k+1}), G(y_k)\rangle - \frac{b_k \eta_k}{\beta_k}\langle G(x_k), G(y_k)\rangle \\
+ c_k L^2 \|x_k - y_{k-1}\|^2 - c_{k+1}L^2\|x_{k+1} - y_k\|^2.
\end{aligned}
\tag{88}
$$

Using Lipschitz continuity and Eq. (86), we have

$$
\begin{aligned}
\|G(x_{k+1}) - G(y_k)\|^2 + \theta L^2\|x_{k+1} - y_k\|^2 &\leq (1+\theta)L^2\|x_{k+1} - y_k\|^2 \\
&= (1+\theta)L^2\|-\eta_k G(y_k)\|^2.
\end{aligned}
\tag{89}
$$

Expanding the left-hand side

$$
\|G(x_{k+1}) - G(y_k)\|^2 + \theta L^2\|x_{k+1} - y_k\|^2 = \|G(x_{k+1})\|^2 - 2\langle G(x_{k+1}), G(y_k)\rangle + \|G(y_k)\|^2 + \theta L^2\|x_{k+1} - y_k\|^2,
$$

and setting $M := L^2(1+\theta)$, inequality (89) is equivalent to

$$
\|G(x_{k+1})\|^2 - 2\langle G(x_{k+1}), G(y_k)\rangle + \left(1 - M\eta_k^2\right)\|G(y_k)\|^2 + \theta L^2\|x_{k+1} - y_k\|^2 \leq 0.
\tag{90}
$$

Recall

$$
\begin{aligned}
V_k - V_{k+1} \geq &\; a_k\|G(x_k)\|^2 - a_{k+1}\|G(x_{k+1})\|^2 + \frac{b_k \eta_k}{\beta_k(1-\beta_k)}\langle G(x_{k+1}), G(y_k)\rangle \\
&- \frac{b_k \eta_k}{\beta_k}\langle G(x_k), G(y_k)\rangle + c_k L^2\|x_k - y_{k-1}\|^2 - c_{k+1}L^2\|x_{k+1} - y_k\|^2.
\end{aligned}
\tag{91}
$$

Multiply (90) by $\frac{b_k \eta_k}{2\beta_k(1-\beta_k)} > 0$ and add the result to the right-hand side of (91) to obtain:

$$
\begin{aligned}
V_k - V_{k+1} \geq &\; a_k\|G(x_k)\|^2 \\
&+ \left(\frac{b_k \eta_k}{2\beta_k(1-\beta_k)} - a_{k+1}\right)\|G(x_{k+1})\|^2 \\
&- \frac{b_k \eta_k}{\beta_k}\langle G(x_k), G(y_k)\rangle \\
&+ \frac{b_k \eta_k}{2\beta_k(1-\beta_k)}\left(1 - M\eta_k^2\right)\|G(y_k)\|^2 \\
&+ L^2\left[c_k\|x_k - y_{k-1}\|^2 + \left(\frac{\theta b_k \eta_k}{2\beta_k(1-\beta_k)} - c_{k+1}\right)\|x_{k+1} - y_k\|^2\right].
\end{aligned}
\tag{92}
$$

We set $a_{k+1} = \frac{b_k \eta_k}{2\beta_k(1-\beta_k)}$, also we had $b_{k+1} = \frac{b_k}{1-\beta_k}$ Define the $2 \times 2$ quadratic form (on $\|G(x_k)\|, \|G(y_k)\|$)

$$
S_k = \begin{pmatrix} S_k^{11} & S_k^{12} \\ S_k^{12} & S_k^{22} \end{pmatrix}
$$

$$
S_k^{11} = a_k, \qquad S_k^{12} = -\frac{b_k \eta_k}{2\beta_k}, \qquad S_k^{22} = \frac{b_k \eta_k}{2\beta_k(1-\beta_k)}\left(1 - M\eta_k^2\right).
$$

Thus, sufficient conditions for $V_k - V_{k+1} \geq 0$ are:

$$
\text{(i)} \quad S_k \succeq 0 \iff S_k^{11} \geq 0,\; S_k^{22} \geq 0,\; S_k^{11}S_k^{22} \geq (S_k^{12})^2,
$$

$$
\text{(ii)} \quad \frac{\theta b_k \eta_k}{2\beta_k(1-\beta_k)} \geq c_{k+1} \geq 0.
$$

We set $\eta_k = c\sqrt{\frac{\beta_k}{M}}$ with $\beta_k = \frac{a}{k+b}$. We choose and check if all conditions work:

$$
S_k^{11} = a_k, \qquad S_k^{12} = -\frac{b_k c}{2\sqrt{M \beta_k}}, \qquad S_k^{22} = \frac{b_k c}{2\sqrt{M \beta_k}(1-\beta_k)}\left(1 - c^2\beta_k\right).
$$

**Condition (i)**   For $S_k^{22} \geq 0$, it is sufficient that

$$a < b, \qquad 0 < c \leq \sqrt{\frac{b}{a}},$$

so that $\beta_k < 1$ and $1 - c^2\beta_k \geq 0$ for all $k$.

Define

$$\frac{(S_k^{12})^2}{S_k^{22}} = \frac{b_k c}{2\sqrt{M}\,\beta_k} \cdot \frac{1 - \beta_k}{1 - c^2\beta_k}, \qquad a_k = \frac{b_{k-1} c}{2\sqrt{M\beta_{k-1}}(1 - \beta_{k-1})}.$$

Condition (i) requires

$$a_k \geq \frac{(S_k^{12})^2}{S_k^{22}}, \qquad \text{for all } k \geq 0.$$

canceling $c, M > 0$ and using $\frac{b_k}{b_{k-1}} = \frac{1}{1 - \beta_{k-1}}$ gives

$$\sqrt{\frac{\beta_k}{\beta_{k-1}}} \geq \frac{1 - \beta_k}{1 - c^2\beta_k}. \tag{*}$$

We assume $b > a$ and

$$0 < c \leq \sqrt{1 - \frac{1}{2a}} \qquad \text{(note that } c < 1\text{)}.$$

Since $0 < c^2\beta_k \leq \beta_k\left(1 - \frac{1}{2a}\right)$, we have

$$1 - c^2\beta_k \geq 1 - \beta_k\left(1 - \frac{1}{2a}\right) = 1 - \beta_k + \frac{\beta_k}{2a},$$

and hence

$$\frac{1 - \beta_k}{1 - c^2\beta_k} \leq \frac{1 - \beta_k}{1 - \beta_k + \frac{\beta_k}{2a}}.$$

With $\beta_k = \frac{a}{k+b}$ and writing $x := k + b$, this becomes

$$\frac{1 - \beta_k}{1 - c^2\beta_k} \leq \frac{\frac{x-a}{x}}{\frac{x-a+\frac{1}{2}}{x}} = \frac{x - a}{x - a + \frac{1}{2}}. \tag{1}$$

Using (1), it suffices to prove

$$\sqrt{\frac{\beta_k}{\beta_{k-1}}} = \sqrt{\frac{x - 1}{x}} \geq \frac{x - a}{x - a + \frac{1}{2}}, \qquad x = k + b \geq b > a. \tag{**}$$

Squaring $(**)$ and cross-multiplying (all terms are positive) is equivalent to

$$(x - 1)\left(x - a + \tfrac{1}{2}\right)^2 \geq x(x - a)^2.$$

Expanding both sides yields the linear inequality

$$x\left(a - \tfrac{3}{4}\right) + \left(a - a^2 - \tfrac{1}{4}\right) \geq 0. \tag{***}$$

If $a \geq 1$ and $b \geq 2$ this holds. We stick to the choice of $a = 1, b = 2$ from this point. This verifies the determinant condition $S_k^{11} S_k^{22} \geq (S_k^{12})^2$ for every $k \geq 1$, where $a_k = \frac{b_{k-1} c}{2\sqrt{M\beta_{k-1}}(1 - \beta_{k-1})} > 0$. The boundary step $k = 0$ is excluded: there $a_0 = 0$, so $S_0^{11} = 0$ while $S_0^{12} \neq 0$, and the determinant condition cannot hold. (Note the formula for $a_0$ would require $\beta_{-1} = 1$, i.e. division by zero, so $a_0$ is not fixed by the recursion.) We handle $k = 0$ separately by telescoping from $k = 1$.

**Condition (ii)** If we set $c_k = a_k$, then using

$$a_{k+1} = \frac{b_k \eta_k}{2\beta_k(1 - \beta_k)}.$$

and choosing $\theta = 1$, the condition (ii) will be satisfied.

Let $H_k := a_k\|G(x_k)\|^2 + b_k\langle G(x_k), x_k - x_0\rangle$. We rewrite $x_k - x_0 = (x_k - x^\star) + (x^\star - x_0)$. By monotonicity of $G$ and the fact that $G(x^\star) = 0$,

$$\langle G(x_k), x_k - x^\star\rangle \geq 0.$$

Hence,

$$H_k \geq a_k\|G(x_k)\|^2 + b_k\langle G(x_k), x^\star - x_0\rangle$$

$$\geq a_k\|G(x_k)\|^2 - \frac{a_k}{2}\|G(x_k)\|^2 - \frac{b_k^2}{2a_k}\|x_0 - x^\star\|^2$$

$$= \frac{a_k}{2}\|G(x_k)\|^2 - \frac{b_k^2}{2a_k}\|x_0 - x^\star\|^2,$$

where the second inequality follows from Young's inequality. Therefore,

$$\frac{a_k}{2}\|G(x_k)\|^2 \;\leq\; H_k + \frac{b_k^2}{2a_k}\|x_0 - x^\star\|^2 \;\leq\; V_k + \frac{b_k^2}{2a_k}\|x_0 - x^\star\|^2.$$

The PSD condition (i) requires $S_k^{11} = a_k > 0$, which holds for every $k \geq 1$ but fails at $k = 0$ (where $a_0 = 0$). We therefore telescope the one-step decrease from $k = 1$: $V_{k+1} \leq V_k$ for all $k \geq 1$, so $V_k \leq V_1$ for all $k \geq 1$. (Starting the telescoping at $k = 1$ sidesteps the $a_0 = 0$ boundary step entirely, exactly as in the proof of Lemma 3.) Combining $V_k \leq V_1$ with $\frac{a_k}{2}\|G(x_k)\|^2 \leq V_k + \frac{b_k^2}{2a_k}\|x_0 - x^\star\|^2$ and dividing by $a_k/2$ gives, for all $k \geq 1$,

$$\|G(x_k)\|^2 \;\leq\; \frac{2V_1}{a_k} + 2\Big(\frac{b_k}{a_k}\Big)^2\|x_0 - x^\star\|^2. \tag{93}$$

With $\beta_k = \frac{1}{k+2}$, we have $1 - \beta_k = \frac{k+1}{k+2}$, and the recursion $b_{k+1} = b_k/(1 - \beta_k)$ telescopes to

$$b_k = b_0 \prod_{t=0}^{k-1} \frac{1}{1 - \beta_t} = b_0 \prod_{t=0}^{k-1} \frac{t+2}{t+1} = b_0\,(k+1).$$

Substituting into $a_{k+1} = \frac{b_k \eta_k}{2\beta_k(1-\beta_k)}$ with $\eta_k = \frac{c}{L}\sqrt{\frac{\beta_k}{2}}$ yields

$$a_{k+1} = \frac{b_0 c}{2\sqrt{2}\,L}\,(k+2)^{3/2}, \qquad \text{i.e.} \qquad a_k = \frac{b_0 c}{2\sqrt{2}\,L}\,(k+1)^{3/2} \quad (k \geq 1).$$

Hence

$$2\Big(\frac{b_k}{a_k}\Big)^2 = \frac{16L^2}{c^2\,(k+1)}, \qquad \frac{a_1}{a_k} = \frac{2\sqrt{2}}{(k+1)^{3/2}}.$$

It remains to bound $V_1$. Since $\beta_0 = \frac{1}{2}$ and $y_0 = x_0$, we have $x_1 = x_0 - \eta_0 G(x_0)$ with $L\eta_0 = \frac{c}{2} \leq 1$, so Lipschitzness gives $\|G(x_1)\| \leq (1 + \frac{c}{2})\|G(x_0)\|$ and $\|x_1 - y_0\| = \eta_0\|G(x_0)\|$. Using $c_1 = a_1$, the identity $b_1\eta_0 = a_1$, $L^2\eta_0^2 = \frac{c^2}{4}$, and $\|G(x_0)\| \leq L\|x_0 - x^\star\|$,

$$V_1 = a_1\|G(x_1)\|^2 + b_1\langle G(x_1), x_1 - x_0\rangle + a_1 L^2\|x_1 - x_0\|^2 \leq a_1\Big[(1+\tfrac{c}{2})^2 + (1+\tfrac{c}{2}) + \tfrac{c^2}{4}\Big]L^2\|x_0 - x^\star\|^2 \leq 4\,a_1 L^2\|x_0 - x^\star\|^2,$$

the last step using $c \leq 1$. Therefore

$$\frac{2V_1}{a_k} \leq 8L^2\,\frac{a_1}{a_k}\,\|x_0 - x^\star\|^2 = \frac{16\sqrt{2}\,L^2}{(k+1)^{3/2}}\|x_0 - x^\star\|^2,$$

and substituting into (93),

$$\|G(x_k)\|^2 \leq \frac{16\sqrt{2}\,L^2}{(k+1)^{3/2}}\|x_0 - x^\star\|^2 + \frac{16\,L^2}{c^2\,(k+1)}\|x_0 - x^\star\|^2,$$

which is the claimed bound with $C_{\text{init}} = 16\sqrt{2}$ and $C_\star = 16/c^2$. Taking the largest step $c = \frac{1}{\sqrt{2}}$ gives

$$\|G(x_k)\|^2 \leq \frac{16\sqrt{2}\,L^2}{(k+1)^{3/2}}\|x_0 - x^\star\|^2 + \frac{32\,L^2}{k+1}\|x_0 - x^\star\|^2,$$

which proves the rate $\|G(x_k)\|^2 = \mathcal{O}((k+1)^{-1})$.

$\square$

## C. Experimental Detail

### C.1. Negative comonotonicity example §6.1

we define $\rho$-*Comonotonicity*, for some $\rho \in \left(-\frac{1}{2L}, \infty\right)$ the field $F$ satisfies

$$\langle Fz - Fz',\, z - z' \rangle \geq \rho\|Fz - Fz'\|^2, \qquad \forall\, z, z' \in \mathbb{R}^d.$$

where L is the lipschitz constant. $\rho < 0$ is referred to as *negative comonotonicity*.

We demonstrate the behavior of (GOMA) with $\eta_k = 0.2$, $\gamma_k = 0.8(1 - \beta_k)$, and $\beta_k = \frac{2}{k+6}$ on the following two–player quadratic saddle game with $L$-Lipschitz and $\rho$-comonotone gradient:

$$f(x, y) = \frac{\rho L^2}{2} x^2 + L\sqrt{1 - \rho^2 L^2}\, xy - \frac{\rho L^2}{2} y^2,$$

The particular instance used in our experiment 1 specializes the parameters to

$$\rho = -\tfrac{1}{3}, \qquad L = 1,$$

so it becomes

$$f(x, y) = -\tfrac{1}{6}x^2 + \tfrac{2\sqrt{2}}{3}xy + \tfrac{1}{6}y^2$$

which is negative comonotonicity ( $\rho < 0$ ), hence it lies outside the scope of our theory (in particular, the comonotonicity/monotonicity assumptions do not cover $\rho < 0$).

As a result, the $O(1/k^2)$-like decay observed for GOMA on this benchmark is purely empirical: we do not claim a guarantee and do not provide a proof in this regime. Any advantage over GOMA in this experiment should be interpreted as evidence of practical robustness rather than a theoretical rate. Formal analysis of GOMA under negative comonotonicity is left as an open problem.

### C.2. Stochastic example §6.2

**Comparison with DSEG**

*Notation.* For consistency with (Hsieh et al., 2020), this subsection adopts their notation: $X_t$ for the iterate, $V$ for the operator, $\hat{V}_t = V(X_t) + Z_t$ for the stochastic oracle, $\mathcal{F}_t$ for the natural filtration, $\beta$ for the Lipschitz constant, and $\gamma_t, \eta_t$ for the DSEG exploration and update step sizes. We write $\kappa_{\text{H}}$ for the noise-growth parameter from their Assumption ($\mathbb{E}[\|Z_t\|^2 \mid \mathcal{F}_t] \leq (\sigma + \kappa_{\text{H}}\|X_t - x^\star\|)^2$), which is distinct from the $\kappa$ in our Assumption 3.

Using the analysis of DSEG (Hsieh et al., 2020), we can see why the method only guarantees convergence to an arbitrarily small neighborhood for general monotone problems.

Their main recursion implies

$$\mathbb{E}\big[\|X_{t+1} - x^\star\|^2 \mid \mathcal{F}_t\big] \leq (1 + C_t\kappa_{\text{H}}^2)\|X_t - x^\star\|^2 - \gamma_t\eta_t\big(1 - \gamma_t^2 L^2 - 8\gamma_t\eta_t\kappa_{\text{H}}^2\big)\|V(X_t)\|^2 + C_t\sigma^2, \tag{94}$$

where $C_t$ depends on $\gamma_t, \eta_t$ and problem constants.

Under their error bound assumption,

$$\|V(x)\| \geq \tau \operatorname{dist}(x, X^\star),$$

a sufficient condition for contraction is

$$\gamma_t \eta_t \left(1 - \gamma_t^2 L^2 - 8\gamma_t \eta_t \kappa_H^2\right) \tau^2 > C_t \kappa_H^2. \tag{95}$$

Since $1 - \gamma_t^2 L^2 - 8\gamma_t \eta_t \kappa_H^2 \leq 1$ and $C_t \geq 4\eta_t^2$, a necessary condition for (95) is

$$\frac{\gamma_t}{\eta_t} > \frac{4\kappa_H^2}{\tau^2}. \tag{96}$$

In their implementation, the step sizes are chosen as $\gamma_t = \alpha/\beta_{\mathrm{alg}}$ and $\eta_t = \alpha$, where $\beta_{\mathrm{alg}} \geq 1$ is a fixed step-size ratio. This yields

$$\frac{\gamma_t}{\eta_t} = \frac{1}{\beta_{\mathrm{alg}}} \leq 1,$$

which contradicts the necessary condition (96) whenever $4\kappa_H^2/\tau^2 > 1$.

In high-dimensional games, $\tau \ll 1$ is typically very small and $\kappa_H = 1 - \frac{1}{n} \approx 1$, so that $4\kappa_H^2/\tau^2 \gg 1$. Consequently, the contraction condition can only be satisfied for vanishingly small step sizes, which leads to an arbitrarily slow convergence rate for DSEG (see Fig. 2).

### C.3. Details of experiments, bilinear game ($\kappa = 1$)

For fair comparison, we performed grid search over key hyperparameters for each method. RAIN (Chen & Luo, 2024) is used in Case I, where its bounded-variance assumption holds; RAIN++(Chen & Luo, 2024) is used in Case II, where neither method is formally in scope but RAIN++'s additional inner prox step empirically offers better robustness. For GOMA, we search over $\eta_{\mathrm{coef}} \in \{0.01, 0.1, 0.3, 0.5\}$ in $\eta_t = \eta_{\mathrm{coef}}\sqrt{\beta_t}$, selecting $\eta_{\mathrm{coef}} = 0.3$. For E-Halpern, we tune the step size factor $\eta_0 \in \{0.5, 1.0, 1.5, 2.0\}$ and batch size cap $B \in \{100, 200, 500\}$, selecting $\eta_0 = 1.5$ and $B = 500$. For RAIN, we search $\tau \in \{0.05, 0.1, 0.2\}$, $\lambda \in \{0.5, 0.7, 0.9\}$, and $\gamma \in \{0.0005, 0.001, 0.005\}$, selecting $\tau = 0.1, \lambda = 0.7, \gamma = 0.001$. For DSEG and FEG , we apply their recommended hyperparameters. For Nesterov, we search the step size $\alpha$ and constant momentum $\beta$ over a $10 \times 11$ grid (including negative momentum), selecting $\alpha = 0.005, \ \beta = -0.1$.

### C.4. Details of experiments, finite-sum game ($\kappa > 1$)

All methods are tuned via grid search to minimize the final $\|F(z_k)\|^2$.

**GOMA.** $\beta_k = 1/(k+2), \eta_k = c\sqrt{\beta_k}$ with $c \in \{1.0, 1.2, 1.5, 2.0, 2.3, 2.5\}$. Best: $c = 1.5$.

**DSEG.** $\eta_k^{(1)} = 0.1/(k+1)^{0.1}, \eta_k^{(2)} = 0.1/(k+1)^{0.9}$. according to (Hsieh et al., 2020)

**FEG.** $\alpha = 1, \rho = 0, \beta_k = 1/(k+1)$. according to (Lee & Kim, 2021)

**RAIN++.** $\mu \in \{0.5, 1, 2, 4\}, \tau \in \{0.05, 0.1, 0.2, 0.3, 0.5\}, \alpha \in \{0.1, 0.2, 0.4, 0.6\}, \gamma \in \{10^{-4}, 10^{-3}, 10^{-2}, 10^{-1}\}$, with 3 inner steps. Best: $(\mu, \tau, \alpha, \gamma) = (0.5, 0.1, 0.1, 10^{-4})$.

**E-Halpern with PAGE.** $\lambda_k = 1/(k+1), p_k = 2/(k+1), s_1 = 6$. Search: $\eta_0 \in \{0.01, 0.05, 0.1, 0.2, 0.5\}, M \in \{1, 3, 9, 27\} \cdot L^2, s_2 \in \{1, 2\}$. Best: $(\eta_0, M, s_1, s_2) = (0.1, 27L^2, 6, 1)$.

**Nesterov.** $y_k = z_k + \beta(z_k - z_{k-1}), z_{k+1} = y_k - \alpha\widehat{F}(y_k)$, with $\alpha \in \{0.01, \ldots, 3.0\}$ and $\beta \in \{-0.9, \ldots, 0.9\}$. Best: $(\alpha, \beta) = (0.01, -0.9)$

## C.5. Extra experiment: Monotone QP Lagrangian

**Setup.** We consider the Lagrangian of a linearly constrained quadratic minimization problem from (Yoon & Ryu, 2021):

$$L(\mathbf{x}, \mathbf{y}) = \tfrac{1}{2}\mathbf{x}^\top \mathbf{H}\mathbf{x} - \mathbf{h}^\top \mathbf{x} - \langle \mathbf{A}\mathbf{x} - \mathbf{b}, \mathbf{y}\rangle, \tag{97}$$

where $\mathbf{x}, \mathbf{y} \in \mathbb{R}^n$, $\mathbf{A} \in \mathbb{R}^{n \times n}$, $\mathbf{b} \in \mathbb{R}^n$, $\mathbf{H} \in \mathbb{R}^{n \times n}$ is positive semidefinite, and $\mathbf{h} \in \mathbb{R}^n$. This saddle function is convex–concave and smooth (due to the quadratic term $\tfrac{1}{2}\mathbf{x}^\top \mathbf{H}\mathbf{x}$); its saddle operator is monotone and 1-Lipschitz.

We use the specific construction from (Yoon & Ryu, 2021):

$$\mathbf{A} = \frac{1}{4}\begin{bmatrix} -1 & 1 & & \\ & \ddots & \ddots & \\ & & -1 & 1 \\ & & & 1 \end{bmatrix} \in \mathbb{R}^{n \times n}, \quad \mathbf{b} = \frac{1}{4}\begin{bmatrix} 1 \\ 1 \\ \vdots \\ 1 \\ 1 \end{bmatrix} \in \mathbb{R}^n, \quad \mathbf{h} = \frac{1}{4}\begin{bmatrix} 0 \\ 0 \\ \vdots \\ 0 \\ 1 \end{bmatrix} \in \mathbb{R}^n, \tag{98}$$

and $\mathbf{H} = 2\mathbf{A}^\top \mathbf{A}$. (Yoon & Ryu, 2021) shows that $\|\mathbf{A}\| \leq \tfrac{1}{2}$, which implies $\|\mathbf{H}\| \leq \tfrac{1}{2}$. Therefore this is a 1-smooth saddle problem. We set $n = 200$.

**Methods.** We compare the same set of methods as in the negative comonotone experiments.

**Hyperparameter selection.**

- **EAG-C / EAG-V**: Since this experiment is from (Yoon & Ryu, 2021), we use their reported parameters: $\alpha = 0.1265/L$ for EAG-C and $\alpha_0 = 0.618/L$ for EAG-V, with $\beta_k = 1/(k+2)$.

- **FEG** (Lee & Kim, 2021): Following the FEG paper, we set $\alpha = 1/L$ with $\beta_k = 1/(k+1)$, which is the same choice used in both monotone and negative comonotone settings.

- **Anchored Popov**: $\alpha = 0.9/L$ with $\beta_k = 1/(k+1)$, selected via grid search over $\alpha = c/L$ with $c \in \{0.3, 0.4, 0.5, 0.618, 0.7, 0.8, 0.9, 1.0, 1.1, 1.2\}$.

- **EG**: $\alpha = 0.5/L$, the same as in (Yoon & Ryu, 2021).

- **DSEG** (Hsieh et al., 2020): Since DSEG is primarily designed for the stochastic setting, we use the same parameters as in our negative comonotone experiments, following the setting of the FEG paper.

- **Nesterov**: $y_k = z_k + \beta(z_k - z_{k-1})$, $z_{k+1} = y_k - \alpha F(y_k)$, selected via grid search over $\alpha \in \{0.01, 0.05, 0.1, 0.3, 0.5, 0.7, 1.0, 1.3, 1.5, 2.0\}/L$ and constant momentum $\beta \in \{-0.9, -0.7, \ldots, 0.9\}$ (including negative momentum). Best: $\alpha = 1/L$, $\beta = -0.1$. The scheduled momentum $\beta_k = k/(k+3)$ diverged at every step size tested.

- **GOMA**: $\alpha = 1.25/L$ with $\eta_k = \alpha$, $\gamma_k = \alpha(1 - \beta_k)$, and $\beta_k = 1/(k+1)$, selected via grid search over $\alpha \in \{0.3, 0.4, 0.5, 0.618, 0.7, 0.8, 0.9, 1.0, 1.1, 1.2, 1.25, 1.3\}/L$.

**Results.** As shown in Figure 3, all anchoring-based methods achieve the optimal $O(1/k^2)$ rate, clearly separating from EG, DSEG, and Nesterov. GOMA outperforms the two-gradient-call methods (EAG-C, EAG-V, FEG) by a constant factor, which is expected given its single gradient call per iteration. Compared to Anchored Popov (which also uses a single call), GOMA achieves a better constant thanks to its two-time-scale structure that decouples the exploration and update step sizes.

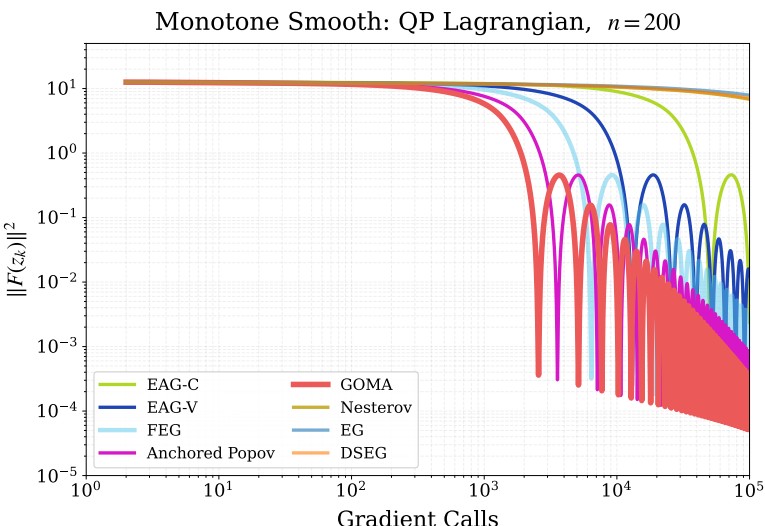

**Figure 3.** Monotone QP Lagrangian ($n = 200$, $L = 1$). Anchoring-based methods (EAG-C/V, FEG, Anchored Popov, GOMA) achieve the $O(1/k^2)$ rate on $\|F(z_k)\|^2$, while EG, DSEG, and Nesterov exhibit a substantially slower decay and largely overlap near the top of the plot. GOMA (red) attains the lowest final $\|F(z_k)\|^2$.

