# OpenReview forum: "Accelerated and Stable Convergence with Anchored Generalized Optimistic Method"
_ICML.cc/2026/Conference — ICML 2026 regular_

### Official Review · Reviewer_8Prf · 2026-03-09

**Soundness:** 3
**Presentation:** 3
**Significance:** 2
**Originality:** 2
**Overall Recommendation:** 4
**Confidence:** 3

**Summary:**

This paper studies first-order methods for unconstrained VI problems, with a particular focus on last-iterate accelerated convergence. The original extragradient methods leverage (i) two time-scale methods for last-iterate convergence and (ii) anchoring for acceleration, but require two gradient evaluations per iteration. This paper designs the GOMA framework based on optimistic methods, which reduces the per-iteration cost to a single gradient query while incorporating two time-scale methods and anchoring, thus achieving last-iterate accelerated convergence. The method is then applied to the stochastic setting.

**Compliance With Llm Reviewing Policy:**

Affirmed.

**Final Justification:**

The authors have resolved my main concerns and I have adjusted my score accordingly.

**Key Questions For Authors:**

**Discussion on Extragradient and Optimistic Methods**

Since this paper already considers both extragradient and optimistic methods, what is the essential technical relationship between them? A clearer discussion of this relationship could be more significant.



**Another Line of Research on Optimization**

The problem of last-iterate acceleration for unconstrained VI seems to have an equivalent form in convex and smooth optimization. Specifically, you can integrate $G(x)$ to obtain a convex and $L$-smooth function $f(x) = \int_{x_0}^x G(x)dx$, and use the inequality $\\|G(x)\\|^2 = \\|\nabla f(x)\\|^2 \le 2L(f(x) - f(x^*))$ to convert the problem into a convergence rate in optimization. A convergence rate of $O(1/T^2)$ would then imply the desired convergence rate for the VI problem. There have been several works in this field worth considering:

- [1] A work that first combines an online *optimistic* algorithm with anytime online-to-batch conversion to achieve an $O(1/T^2)$ convergence rate.

- [2] A work that obtains an $O(1/T^2)$ convergence rate and requires only one gradient query per iteration.

Therefore, I’m curious whether these two fields share some essential commonality.



[1] UniXGrad: A universal, adaptive algorithm with optimal guarantees for constrained optimization, NeurIPS 2019

[2] A Simpler Approach to Accelerated Stochastic Optimization: Iterative Averaging Meets Optimism, ICML 2020

**Limitations:**

Yes.

**Strengths And Weaknesses:**

**Strengths:** This paper is well-structured in reviewing existing methods and positioning its contribution within the research. The proposed algorithmic framework clearly introduces its relationship to the original methods, and it is flexible with strong theoretical guarantees.

**Weaknesses:**

- The paper provides limited discussion on the technical challenges of using optimistic updates instead of extragradient with two time-scales and anchoring. This may leave readers uncertain about whether there are essential difficulties in analyzing the method, especially since the two methods seem to share many similarities.
- Regarding the main motivation for studying optimistic methods, that is, reducing the per-iteration cost to a single gradient query, the paper ultimately focuses on the asymptotic convergence rate. However, the concept of "iteration" is somewhat artificially defined by the algorithm. As a result, the discussion on how much benefit this reduction in cost provides is limited.

---

> ### Author Rebuttal · Authors · 2026-03-31
>
> We thank Reviewer 8Prf for the positive assessment of the paper's structure and framework. However, we respectfully note that the review contains several factual inaccuracies that may have led to an underestimation of our contribution. We address these below and hope the clarifications will lead the reviewer to reconsider their assessment.
>
> ## TL;DR
> - **Important update:** Since submission, we have obtained a new proof of Theorem 4 achieving last-iterate convergence with a *constant* batch size. See our response to Reviewer mWtv (Q2) for details.
> - **Factual correction:** The reviewer's summary misattributes two-time-scale and anchoring to extragradient methods; neither is a feature of classical EG. We clarify the precise lineage.
> - **EG vs. optimistic relationship:** As discussed in Section 3.1, the single-query design of optimistic methods is not only a computational convenience but a requirement for compatibility with the online learning model, where only one gradient per round is accessible.
> - **Convex optimization connection:** The suggested reduction applies only when $G$ is a gradient field, which excludes the game-theoretic settings central to our paper.
>
>
> ## On the Summary and EG/OG Relationship (W1 & Q1)
>
> The reviewer writes:
>
> > *"The original extragradient methods leverage (i) two time-scale methods for last-iterate convergence and (ii) anchoring for acceleration, but require two gradient evaluations per iteration."*
>
> This is incorrect. Classical EG (Korpelevich, 1976) uses **neither** two time-scale step sizes **nor** anchoring:
> $$y_k = x_k - \eta_k G(x_{k-1}), \quad x_{k+1} = x_k - \eta_k G(y_k). \tag{EG}$$
> Two-time-scale methods were introduced later by Hsieh et al. (2020) to stabilize stochastic extragradient dynamics, and anchoring was introduced separately (Diakonikolas, 2020; Yoon & Ryu, 2021). GOMA is the first to combine all three within a single-call optimistic framework, and we believe this mischaracterization may have contributed to an underestimation of our contribution.
>
> Regarding the EG/OG relationship as discussed in Section 3.1, in the online learning model, only one gradient per round is available (Golowich et al., 2020), making the second EG query structurally inaccessible. Hsieh et al. (2020) further show that standard EG fails in stochastic bilinear games and that two-time-scale updates are necessary for convergence. GOMA addresses both issues simultaneously, achieving $\mathcal{O}(1/k^2)$ last-iterate convergence and provable stochastic convergence where EG fails.
>
> Golowich, N., Pattathil, S., and Daskalakis, C. Tight last-iterate convergence rates for no-regret learning in multi-player games. NeurIPS, 2020.
>
> Hsieh, Y.-G., Iutzeler, F., Malick, J., and Mertikopoulos, P. Explore aggressively, update conservatively: Stochastic extragradient methods with variable step-size scaling. NeurIPS, 2020.
>
> ## On the Connection to Convex Optimization (Q2)
>
> The reviewer writes:
>
> > *"You can integrate $G(x)$ to obtain a convex and $L$-smooth function $f(x) = \int_{x_0}^{x} G(x)\,dx$, and use the inequality $\|G(x)\|^2 = \|\nabla f(x)\|^2 \leq 2L(f(x) - f(x^*))$ to convert the problem into a convergence rate in optimization."*
>
> This reduction is not applicable to the problems we study. A monotone operator $G$ can be written as a gradient field $\nabla f$ if and only if $G$ is *conservative* (i.e., its Jacobian is symmetric). This is a standard structural distinction between optimization and variational inequalities (see, e.g., Facchinei & Pang, 2003). For instance, the bilinear problem discussed in Section 4.2 has $G(x,y) = (Ly, -Lx)$, whose Jacobian is skew-symmetric and is therefore not a gradient field. No convex function $f$ exists such that $G = \nabla f$ for these problems, so the integral $\int G(x)\,dx$ is not well-defined.
>
> This is precisely why the VI literature requires dedicated algorithmic tools rather than reducing to convex optimization. The references the reviewer cites work within the convex optimization setting:
>
> - **[1] (Kavis et al., 2019):** Studies online *convex optimization* (where $G = \nabla f$), and provides *averaged-iterate* convergence via online-to-batch conversion. This does not yield last-iterate guarantees for non-conservative monotone operators.
> - **[2] (Joulani et al., ICML 2020):** Studies *smooth convex optimization* and achieves $\mathcal{O}(1/T^2)$ convergence for *function values* via iterative averaging combined with optimism. This again requires $G = \nabla f$ and provides averaged-iterate, not last-iterate, convergence.
>
> In summary, these works address a different problem class. Our results hold for **all** monotone Lipschitz operators, a strictly broader class that includes games and saddle-point problems where $G$ has no underlying scalar objective (Gidel et al., 2019).
>
> Gidel, G., Berard, H., Vignoud, G., Vincent, P., and Lacoste-Julien, S. A Variational Inequality Perspective on Generative Adversarial Networks. ICLR, 2019.

---

> > ### Author Rebuttal · Reviewer_8Prf · 2026-04-02
> >
> > I thank the authors for their rebuttal, which clarified several points. They have addressed my main concerns, and I have accordingly raised my score. Although I remain curious about the technical similarities between variational inequalities and optimization in the use of optimism for acceleration, this question is beyond the scope of the paper.

---

### Official Review · Reviewer_LYcf · 2026-03-11

**Soundness:** 3
**Presentation:** 3
**Significance:** 3
**Originality:** 3
**Overall Recommendation:** 5
**Confidence:** 4

**Summary:**

The paper proposes a generalisation of the optimistic method with two step-size sequences and anchoring, named GOMA, for solving unconstrained monotone smooth variational inequalities. GOMA achieves an optimal $\mathcal{O}(\frac{1}{\sqrt{\epsilon}})$ complexity bound with deterministic oracles and an $\mathcal{O}(\frac{1}{\epsilon^4})$ complexity bound with stochastic oracles and increasing batch sizes. The paper additionally provides experiments supporting the effectiveness of GOMA. The paper is well written and technically solid.

**Compliance With Llm Reviewing Policy:**

Affirmed.

**Final Justification:**

I believe that this paper should be accepted given that the new results on not-growing batch size are correct and added to a final version of the paper. Authors addressed all my concerns and answered al questions.

**Key Questions For Authors:**

Questions:
1. Why was Section 4.2 introduced? Could you please elaborate more in the paper on why this comparison between this bilinear example and general G is needed?
2. The paper does not state what exactly the decrease in the Lyapunov function is: does the statement of Lemma 1 imply that $V_{k+1} < V_{k}$, or does a more strict inequality hold?
3. "Several works have proposed ways to accelerate this rate, including Tran-Dinh & Luo (2021) and Abe et al. (2024)." It would be nice to elaborate on what the results of these works are, on page 2, column 2, lines 060–063.
4. Does GOMA achieve an optimal complexity bound for smooth monotone SVIs? Are there methods that achieve better than $\mathcal{O}(\frac{1}{\epsilon^4})$ complexity under the same assumptions?
5. It seems that the theoretical results for GOMA developed for a version of gradient method evaluated at an anchored point. What are the complexity of this version of GOMA for deterministic VI?

**Limitations:**

The authors adequately discussed the limitations and potential negative societal impact of their work.

**Strengths And Weaknesses:**

**Strengths**
1. The paper provides a good overview and intuition for the developed method.
2. The proposed method recovers optimal complexity bounds for solving deterministic smooth monotone VIs.
3. GOMA shows better performance compared to the previous methods considered in the paper in a negative comonotone game with a deterministic oracle and in bilinear and finite-sum minimax problems with stochastic oracles.

**Weaknesses**
The main weakness of the paper is the lack of a detailed comparison of the proposed GOMA with the existing literature and a slightly unfair comparison of GOMA and existing methods in experiments.
1. On lines 321–324 it is said "Since the stochastic variant in Section 5.2 uses a linearly growing minibatch of size k+1, ..."; there is no statement on how $k$ grows. Please provide it in the statement of Theorem 4. It is an important part of the paper.
2. When comparing the results of Theorem 4, please provide a detailed comparison of the complexity obtained in this paper and in the previous works with all methods.
3. It seems like the problem from Experiment 3.1 does not satisfy the assumptions the paper considers. The operator there is negative-comonotone, while the paper provides theoretical results for monotone operators. It would be beneficial to have a fair comparison of the performance of the proposed method with the previous methods on monotone smooth VIs.
4. I believe it is unfair to compare, for example, EG+ with GOMA without increasing batch sizes as prescribed by theory. EG+ provides convergence rates without increasing batch sizes, thus its stepsizes (of order $\frac{1}{k^3}$) decrease to zero faster than those for GOMA (order of $\frac{1}{\sqrt{k}}$). Restricting EG+ to follow theoretically chosen stepsizes while not following the theory developed in Theorem 4 for GOMA is unfair. Could you please provide experiments where GOMA uses a batch size that follows the Theorem 4 statement?

Minor:
1. The statement of Theorem 1 is confusing. $a_k, b_k, c_k$ are not parameters of the algorithm but constants arising in the analysis. It is advised to rewrite the statement of Theorem 1.
2. In the statement of Theorem 2 (page 4, column 2, lines 205–206) it should be "for any $\gamma^* < \frac{1}{3\sqrt{2}L}$" instead of $\eta^*$.
3. Starting from page 6, column 1, line 315, the paper uses three different notations for iterate indices: N, T, k. Also, before line 315 the index k was the iterate number; starting from Section 5 it is the number of samples in a mini-batch. This notation is overloaded.

---

> ### Author Rebuttal · Authors · 2026-03-31
>
> We thank Reviewer LYcf for the detailed and constructive review, and for recognizing the good overview, optimal complexity bounds, and strong experimental performance of GOMA.
>
> ## TL;DR
> - **W1/W2/W4:** Batch size fully resolved; see mWtv Q2.
> - **W3:** Added monotone smooth VI comparison experiment.
> - **Q1–Q5:** Provided clarifications.
>
> ## Batch-Size Growth in Theorem 4 (W1, W2, W4)
>
> **We are happy to report that this concern is fully resolved.** Since submission, we have obtained a new proof of Theorem 4 that removes the growing batch size assumption entirely. The full proof sketch and new theorem are provided in our response to Reviewer mWtv (Q2). The revised Theorem 4 achieves last-iterate convergence of $\mathbb{E}\|G(x_k)\|^2$ with a constant batch size, which to the best of our knowledge is a new result in this setting.
>
> We will also include a detailed complexity comparison table across methods in the revision, noting the different metrics (gap vs. gradient norm) and settings (bounded vs. unbounded domain) under which each result holds.
>
> This also directly resolves the experimental fairness concern (W4): the new theory now backs the constant-batch-size experiments already reported.
>
> ## Adding a Monotone Smooth VI Comparison Experiment (W3)
>
> We include the negative comonotone experiment to show empirical robustness beyond the monotone setting. We agree a direct monotone comparison is valuable and have added one in the revision.
>
> **Setup.** We use the convex-concave Lagrangian saddle-point problem from Yoon & Ryu (2021) (construction of Ouyang & Xu., $n=200$, monotone $1$-Lipschitz operator). We compare the same methods as in the negative comonotone experiment, with hyperparameters taken from the respective papers or selected via grid search. For GOMA: $\alpha = 1.25/L$, $\eta_k = \alpha$, $\gamma_k = \alpha(1-\beta_k)$, $\beta_k = 1/(k+1)$.
>
> **Results.** All anchoring-based methods achieve the optimal $O(1/k^2)$ rate, separating from EG and EG+. GOMA outperforms two-call methods (EAG-C, EAG-V, FEG) by a constant factor and outperforms Anchored Popov (also single-call) due to its two-time-scale structure.
>
> https://ibb.co/whCqjVm3
>
> ## Purpose of Section 4.2 (Q1)
>
> Section 4.2 analyzes GOMA on the bilinear example to verify that our general bound (Theorem 1) is tight up to constants, since the lower bound is established on the same problem class. We will add a clarifying sentence at the beginning of Section 4.2 in the revision.
>
> ## Lyapunov Function Decrease (Q2)
>
> Yes, a strict decrease holds. Lemma 1 establishes a one-step descent inequality of the form $V_{k+1} \leq V_k - \Phi_k$, where $\Phi_k$ collects squared gradient norms and iterate differences, and is strictly positive away from the solution. Lemma 1 establishes the conditions on the step sizes under which $\Phi_k \geq 0$, and Lemma 2 provides the concrete parameter regime (explicit values of $\eta_*$ or $\gamma_*$ and $\beta_k$) under which these conditions are satisfied. We will make this clear in the revised statement of Lemma 1.
>
> ## Elaboration on Accelerated Methods (Q3)
>
> We note that Tran-Dinh & Luo (2021) propose the Anchored Popov method, which arises as a special case of GOMA with $\gamma_k = \eta_k$ (discussed in Sections 3.3 and 4) and is included in the experiments. Abe et al. (2024) is technically equivalent to the Anchored Popov method. We will add a brief clarifying note on lines 060–063 pointing readers to these discussions.
>
> ## Optimality of GOMA for Stochastic VIs (Q4)
>
> There are indeed relevant lower bounds. Chen & Luo (2022) establish a near-optimal SFO complexity of $\tilde{\Omega}(\sigma^2/\varepsilon^2 + LD/\varepsilon)$ for stochastic convex-concave minimax problems, and their RAIN algorithm matches this up to logarithmic factors. However, their setting differs from ours in important ways: they work on bounded domains (diameter $D$) and rely on recursive variance reduction techniques. GOMA's stochastic variant does not use variance reduction and operates on unbounded domains, which are more challenging conditions. Closing this gap while maintaining GOMA's single-call, variance-reduction-free design is an important open question that we will flag in the revision.
>
> Chen, L. and Luo, L. Near-Optimal Algorithms for Making the Gradient Small in Stochastic Minimax Optimization. JMLR, 2024.
>
> ## Complexity of Anchored GOMA Variant (Q5)
>
> Since all versions of GOMA include anchoring, we believe the reviewer is referring to the stochastic variant ($\diamond$) where $\gamma_k = 0$. We derive the deterministic rate of this variant in Appendix B.2: it achieves $O(1/k)$, which is *not* accelerated. This highlights an important point: to achieve the accelerated $O(1/k^2)$ rate, one must use a proper two-time-scale relationship between $\gamma_k$ and $\eta_k$ as specified in Theorems 1 and 2. We will make sure this point is clearly stated in the main text.

---

> > ### Author Rebuttal · Reviewer_LYcf · 2026-04-04
> >
> > I thank the authors for their detailed response. All my concerns are resolved. I will increase the score.

---

> > > ### Author Response · Authors · 2026-04-04
> > >
> > > Thank you so much for your thoughtful review and for taking the time to engage with our response. We are glad we could address your concerns. We noticed that the score may not have been updated yet on the system, and wanted to flag this in case it was an oversight. Thank you again for your support of our work!

---

### Official Review · Reviewer_Ubjt · 2026-03-13

**Soundness:** 3
**Presentation:** 3
**Significance:** 3
**Originality:** 3
**Overall Recommendation:** 5
**Confidence:** 3

**Summary:**

The paper studies first-order methods for solving monotone variational inequalities in min-max optimization. The proposed algorithm, GOMA, combines two time-scale optimistic updates with an anchoring term from Halpern iteration. The proposed algorithm achieves a last iterate convergence rate of $K(1 / k^2)$, which is the optimal rate.

**Compliance With Llm Reviewing Policy:**

Affirmed.

**Final Justification:**

The authors have resolved my main concerns and I have adjusted my score accordingly.

**Key Questions For Authors:**

See questions above.

**Limitations:**

See questions above.

**Strengths And Weaknesses:**

The paper is well-written, and the key idea is clearly delivered. It is technically solid and contributes to the literature of monotone variational inequality.

My major questions and comments are as below.

1. Choice of the hyperparameter. Section 4.1 discusses the last iterate convergence rates under two different cases, namely a large update regime and a large exploration regime. It is not fully clear to me how to choose between two regimes. As suggested by the authors, "Each setup may be preferable depending on whether a larger exploration step size or a larger update step size is desired." Why does it matter if both choices lead to qualitatively similar convergence rate of $1 / k^2$?

2. Numerical experiments in Section 6.2.1. In the left panel of Figure 2, the GOMA curve appears to exhibit more abrupt changes than several of the baselines. Could the authors provide some intuition for this behavior? Is it related to the choice of hyperparameters / anchoring schedule used for GOMA?

3. Assumptions behind convergence. The paper’s theory is developed for monotone, Lipschitz operators, which is a standard setting for variational inequality methods, but it is also a fairly structured regime. In particular, this limits the direct applicability of the convergence guarantees to more general or complex min-max problems that may fall outside the monotone setting. It would strengthen the paper to clarify which practically relevant classes of min-max problems satisfy monotonicity, and which important settings are excluded.



My minor comments are as below.

1. Please double check the typos in the paper. For instance, two-time-scale and two time-scale both appear in the paper.

---

> ### Author Rebuttal · Authors · 2026-03-31
>
> We thank Reviewer Ubjt for the positive evaluation and for noting that the paper is well-written with clearly delivered key ideas.
>
> ## TL;DR
> - **Important update:** Since submission, we have obtained a new proof of Theorem 4 achieving last-iterate convergence with a *constant* batch size. See our response to Reviewer mWtv (Q2) for details.
> - **Choice between regimes:** We clarify when each regime is preferable and why the distinction matters beyond asymptotic rates.
> - **GOMA curve behavior in Fig. 2:** We explain the oscillatory pattern.
> - **Applicability beyond monotone setting:** We clarify which practical problems are covered and which are excluded.
> - **Typos:** We will fix the inconsistent hyphenation in the revision.
>
> ## Choice of Hyperparameter Regime
>
> The reviewer asks:
>
> > *"It is not fully clear to me how to choose between two regimes... Why does it matter if both choices lead to qualitatively similar convergence rate of $1/k^2$?"*
>
> The $O(1/k^2)$ rate is a worst-case guarantee over the class of monotone $L$-Lipschitz VIs. Two setups with the same worst-case rate can behave quite differently on specific instances within that class. As simple examples: on the 1D quadratic $G(x) = \alpha x$, the large update regime (large $\eta_*$) yields a tighter bound constant since the Lyapunov decrease in Theorem 1 scales as $1/\eta_*$ in that case. On the bilinear game $\min_x \max_y c \cdot xy$, where the operator is purely rotational, the large exploration regime (large $\gamma_*$) better anticipates the changing gradient direction and reduces overshooting. In practice, the best choice depends on the problem structure and is most reliably found via hyperparameter search.
>
> ## Oscillatory Behavior of GOMA in Fig. 2
>
> The reviewer asks:
>
> > *"In the left panel of Figure 2, the GOMA curve appears to exhibit more abrupt changes than several of the baselines. Could the authors provide some intuition for this behavior? Is it related to the choice of hyperparameters / anchoring schedule used for GOMA?"*
>
> The abrupt changes might arise since GOMA uses a single stochastic sample per iteration without any variance reduction. In contrast, E-Halpern employs PAGE variance reduction and RAIN uses multi-step inner loops, both of which naturally smooth the trajectory.
>
>
> ## Applicability Beyond Monotone Setting
>
> The reviewer notes:
>
> > *"It would strengthen the paper to clarify which practically relevant classes of min-max problems satisfy monotonicity, and which important settings are excluded."*
>
> Monotonicity covers a broad range of practically relevant problems, including bilinear saddle-point problems, zero-sum matrix games, convex-concave min-max problems arising in robust optimization and distributionally robust learning, Nash equilibrium computation in multi-agent RL (under linear or convex reward structure), and adversarial training with convex loss functions. It excludes non-convex non-concave problems such as general multi-agent games with nonlinear payoffs and settings satisfying only weak-MVI. That said, GOMA performs well empirically on negative comonotone games (Fig. 2), suggesting broader applicability. A formal weak-MVI extension is an important direction for future work. We will add this discussion to the revision.
>
> ## Typos
>
> We will fix the inconsistent usage of "two-time-scale" vs. "two time-scale" throughout the paper. Thank you for pointing this out.

---

> > ### Author Rebuttal · Reviewer_Ubjt · 2026-04-03
> >
> > My questions are fully addressed.

---

### Official Review · Reviewer_mWtv · 2026-03-13

**Soundness:** 4
**Presentation:** 3
**Significance:** 2
**Originality:** 2
**Overall Recommendation:** 4
**Confidence:** 4

**Summary:**

The paper considers unconstrained smooth and monotone variational inequality problems under full and stochastic information feedback. The authors propose an algorithm that combines Halpern-type anchoring and Popov-style optimism and analyze its finite-iterate and stochastic gradient oracle call complexity. The convergence complexity achieves the lower bound in terms of complexity in the deterministic setting.

In section 4, the authors present Generalized Optimistic Method with Anchoring (GOMA).
$$\begin{align} y_k = \beta_k x_0  + ( 1 - \beta_k ) x_k + \gamma_k G(y_{k-1}), \quad x_{k+1} = \beta_k x_0  + ( 1 - \beta_k ) x_k + \eta_k G(y_{k}) \end{align}$$
GOMA is an optimistic gradient method where the "exploration step" ($\gamma_k$) is of different size to that of the "update step" ($\eta_k$). Additionally, the update is anchored to the initialization $x_0$. Halpern-type anchoring has been analyzed in relevant literature [1,2].

Using a clear Lyapunov/potential functional argument, a rate of convergence $| G(x_k) |^2 = O(1/k^2)$ is proven in Theorem 1. This rate of convergence matches the computational lower bound [3] and previous literature of the same setting [3,4].

In section 5, the authors contribute an algorithm for the setting of stochastic first-order oracle. Similar to section 4, they use a Lyapunov style algorithm to show last-iterate convergence of the expected norm squared of the operator. In this section, GOMA is instantiated with $\gamma_k=0$. This is a single-gradient-oracle-call algorithm with last-iterate convergence guarantees.

The authors go on to analyze particular monotone variational inequality problems and also test the empirical performance of their method in non-monotone problems with commendable success.

[1] Diakonikolas, J. (2020) ‘Halpern iteration for near-optimal and parameter-free monotone inclusion and strong solutions to variational inequalities’. *Proceedings of the Conference on Learning Theory*.

[2] Tran-Dinh, Q. and Luo, Y. (2021) ‘Halpern-type accelerated and splitting algorithms for monotone inclusions’. *arXiv preprint* arXiv:2110.08150.

[3] Yoon, T. and Ryu, E.K. (2021) ‘Accelerated algorithms for smooth convex-concave minimax problems with (O(1/k^2)) rate on squared gradient norm’. *Proceedings of the 38th International Conference on Machine Learning*.

[4] Yoon, T., Kim, J., Suh, J.J. and Ryu, E.K. (2024) ‘Optimal acceleration for minimax and fixed-point problems is not unique’. *arXiv preprint*.

**Compliance With Llm Reviewing Policy:**

Affirmed.

**Key Questions For Authors:**

* What would be the main difficulty in generalizing your results to the weak-MVI case?
* Can the increasing batch size design choice be overcome with a different technique?
* Do you believe that you could get similar results as Theorem 1 and Theorem 4 for the appropriate stationarity proxy in the constrained optimization setting?

**Limitations:**

yes

**Strengths And Weaknesses:**

Strengths:
* The main text does a good in presenting relevant literature.
* The paper is theoretically sound.
* For the deterministic gradient oracle setting:
  * Matching the lower bound with a single-gradient-call method
  * Even though the algorithm has two different step sizes, Theorem 1 allows a more convenient to tune it through a choice of step size coupling.
* For the stochastic gradient oracle setting:
  * The proposed algorithm for this setting is simpler than previously proposed methods with the same iteration complexity.
  * The guarantee of the contributed algorithm is for the expected operator norm squared of the last iterate.


Weaknesses:
* For the deterministic case:
  * The VI problems considered are unconstrained. There already exist algorithms whose iteration complexity upper bound matches the theoretical lower bound.
* For the stochastic case:
  * The algorithm achieves last iterate convergence but with a linearly increasing batch size.
* Proofs in the appendix have no outline. Equations lay next to each without a prior explanation of what particular quantity the authors are trying to control in their potential function.
* Other works already analyze nonmonotone weak-Minty VIs [1] with similar techniques. This paper only considers the rather over-investigated setting of monotone variational inequalities.

---

[1] Diakonikolas, J., Daskalakis, C. and Jordan, M.I., 2021, March. Efficient methods for structured nonconvex-nonconcave min-max optimization. In International Conference on Artificial Intelligence and Statistics (pp. 2746-2754). PMLR.

---

> ### Author Rebuttal · Authors · 2026-03-31
>
> We thank Reviewer mWtv for their careful and thorough review, for the accurate summary of our contributions, and for recognizing the strengths of our work including the matching of the lower bound with a single-gradient-call method and the simpler algorithm design for the stochastic setting.
>
> ## TL;DR
> - **Increasing batch size (Q2):** Fully resolved. New proof of Theorem 4 achieves last-iterate convergence with constant batch size; proof sketch and theorem below.
> - **Weak-MVI (Q1):** No last-iterate result is known; the barrier may be fundamental.
> - **Constrained setting (Q3):** No accelerated last-iterate result exists even for two-call methods; two structural barriers identified.
> - **Proof outlines:** We will add roadmaps to the appendix proofs in the revision.
>
>
> ## Increasing Batch Size (Q2)
>
> The reviewer asks:
>
> > *"Can the increasing batch size design choice be overcome with a different technique?"*
>
> **Yes, and we have done so since submission.** We have obtained a new proof of Theorem 4 that removes the growing batch size assumption entirely, achieving last-iterate convergence of $\mathbb{E}\|G(x_k)\|^2$ with a constant batch size. To the best of our knowledge, this is the first such result for stochastic monotone Lipschitz VIs without growing batch sizes or variance reduction.
>
> **Proof sketch.** The key insight is in Equation (75) of the appendix. Instead of applying the Cauchy-Schwarz inequality, we observe that  $\eta_k(\widehat{G}(y_k,\xi_k)-G(y_k)) = x_{k+1} - \tilde{y}_k$ and use monotonicity of $G$ to show:
>
> $$\mathbb{E}[\langle G(x_{k+1})-G(\tilde{y}_k),\,\widehat{G}(y_k,\xi_k)-G(y_k)\rangle \mid \mathcal{F}_k]$$
>
> $$= -\frac{1}{\eta_k}\mathbb{E}[\langle G(x_{k+1})-G(\tilde{y}_k), x\_{k+1} -\tilde{y}_k\rangle \mid \mathcal{F}_k] \leq 0.$$
>
>
> This leaves only higher-order noise terms in $\eta_k$, yielding a faster rate without growing minibatches.
>
> **New theorem.** Let $G$ be monotone and $L$-Lipschitz, and let $\hat{G}(x,\xi)$ satisfy $\mathbb{E}[\|\hat{G}(x,\xi)\|^2] \leq \sigma^2 + \kappa\|G(x)\|^2$ for $\sigma \geq 0$, $\kappa \geq 1$. With $\beta_k = \frac{2}{k+2}$ and $\eta_k = \frac{2^{3/4}}{L\sqrt{\kappa}(k+2)^{3/4}}$, for all $N \geq 0$:
> $$\mathbb{E}\|G(x_N)\|^2 \leq \frac{2\sqrt{2}L^2\kappa\|x_0-x^*\|^2}{\sqrt{N+2}} + \frac{3\sigma^2\kappa}{4\sqrt{N+2}}.$$
>
> ## Extension to Weak-MVI (Q1)
>
> The reviewer asks:
>
> > *"What would be the main difficulty in generalizing your results to the weak-MVI case?"*
>
> The core difficulty is that, to the best of our knowledge, no last-iterate convergence result exists for weak-MVI, and deriving one may be fundamentally impossible. Our entire analysis is built around last-iterate bounds via the Lyapunov potential in Lemma 1, which critically relies on monotonicity. Extending to weak-MVI would therefore require a different proof strategy and likely a different convergence measure. That said, GOMA empirically converges on negative comonotone games (Fig. 2), suggesting the method may be more broadly applicable than the current theory captures.
>
>
> ## Constrained Setting (Q3)
>
> The reviewer asks:
>
> > *"Do you believe that you could get similar results as Theorem 1 and Theorem 4 for the appropriate stationarity proxy in the constrained optimization setting?"*
>
>
>
> The constrained setting introduces two challenges absent from our analysis. First, the stationarity measure changes: $\|G(x^\star)\| \neq 0$ in general for constrained solutions, so one must use either the gap function or the tangent residual. Last-iterate convergence at the non-accelerated $O(1/N)$ rate has been established for optimistic gradient in the constrained setting (Gorbunov, Taylor, Gidel, NeurIPS 2022), and at $O(1/\sqrt{T})$ using the tangent residual (Cai et al., NeurIPS 2022). No $O(1/k^2)$ result exists in the constrained setting, even for two-call methods. Second, the anchoring step in GOMA conflicts with projection: the pull toward $x_0$ is distorted after projecting onto $\mathcal{X}$, breaking the telescoping structure of our Lyapunov function. Whether accelerated last-iterate convergence is achievable in the constrained setting is an open problem.
>
> Gorbunov, E., Taylor, A., and Gidel, G. Last-iterate convergence of optimistic gradient method for monotone variational inequalities. NeurIPS, 2022.
>
> Cai, Y., Oikonomou, A., and Zheng, W. Tight last-iterate convergence of the extragradient and the optimistic gradient descent-ascent algorithm for constrained monotone variational inequalities. NeurIPS, 2022.
>
> ## Proof Outlines in the Appendix
>
> We acknowledge that the appendix proofs would benefit from more structural commentary. In the revision, we will add a proof roadmap at the beginning of each major proof, explaining the key quantities being controlled in the potential function and the high-level strategy before diving into the algebraic details.
>
> ---

---

> > ### Author Rebuttal · Reviewer_mWtv · 2026-04-06
> >
> > I thank the authors for their answers. They addressed my concerns yet I still believe the results are limited (no MVI convergence results, no constrained optimization, same rates exist for the setting with deterministic feedback).

---

> > > ### Author Response · Authors · 2026-04-08
> > >
> > > We thank the reviewer for taking time to engage with our work and their positive assessment of our paper.
> > >
> > >
> > > **No MVI convergence results / No constrained optimization.** We agree that extending to weak-MVI and constrained settings would be valuable, and we discussed the structural barriers in our initial response (Q1 and Q3). We would like to point out that last-iterate convergence in the weak-MVI case might not be possible to prove, as it remains an open problem for the field, and we are not aware of any analogous results. That said, we might be missing relevant work. **Could the reviewer point to any paper that achieves last-iterate convergence in the weak-MVI case?** To the best of our knowledge, no such result exists.
> > >
> > > **Same rates in the deterministic setting.** The fact that GOMA matches the $O(1/k^2)$ lower bound confirms that it is an optimal algorithm for this problem class. The natural follow-up question is: does it offer any advantage over existing methods? We believe the answer is clearly yes. **No other method can achieve last-iterate convergence in the stochastic setting without growing batch sizes or variance reduction.** Stochastic GOMA is a surprisingly simple algorithm that we believe is both practically and theoretically important, and will find broad use in multi-agent and game-theoretic problems that can be formulated as VIs:
> > > $$x_{k+1} = x_k + \beta_k(x_0 - x_k) - \eta_k G(x_k + \beta_k(x_0 - x_k)). \quad \text{(Eq. 14)}$$
> > > As reported in our initial response (Q2), we have removed the growing batch size requirement entirely, achieving last-iterate convergence with a constant batch size, which to our knowledge is a first for monotone Lipschitz VIs without variance reduction. We believe this substantially strengthens the paper's contribution beyond matching known deterministic rates.
> > >
> > > We thank the reviewer again for the thoughtful engagement and welcome any further questions.

---

### Decision · Program_Chairs · 2026-04-30

**Decision:**

Accept (regular)

**Comment:**

In this paper, the authors study last-iterate convergence when solving monotone variational inequalities with an unconstrained strategy space. They propose Generalized Optimistic Methods with Anchoring (GOMA), which combines two time-scale algorithms accelerated by an anchoring term inspired by Halpern iteration. They demonstrate that one variant of GOMA with a positive exploration step achieves an accelerated convergence rate of $O(1/k^2)$. In the stochastic setting, the authors use a different GOMA variant without an exploration step to achieve a convergence rate of $O(1/\sqrt{k})$.


The reviewers liked the paper's overall results, especially the tightness of the bounds for both settings.
One major drawback of the paper concerns the comparison with related work. For the non-stochastic setting, optimal complexity bounds have already been achieved using a similar acceleration technique (e.g., [1] Theorem 1 and 2), at least the paper and the discussion does not seem to clarify the originality well. For the stochastic setting, the authors claim in the discussion that "this is the first such result for stochastic monotone Lipschitz VIs without growing batch sizes or variance reduction." However, at least in the constrained setting, some work has already been done (e.g., [2] Theorem 4.3), and I don't see evidence that extending to the unconstrained setting would be difficult. Nevertheless, this work would be valuable for the community, so I recommend weak acceptance. The authors are encouraged to incorporate the comments and suggestions in the revision.

The first reference in the paper seems to be an AI hallucination.
Mertikopoulos, P., Papadimitriou, C., and Piliouras, G. Optimistic mirror descent in saddle-point problems: Going
the extra(-gradient) mile. Theoretical Computer Science,
807:442–470, 2020. (Cit. on p. 1)
The correct reference is:
Mertikopoulos, Panayotis, et al. "Optimistic Mirror Descent in Saddle-Point Problems: Going the Extra (Gradient) Mile." ICLR 2019-7th International Conference on Learning Representations. 2019.


[1] Yoon, T. and Ryu, E.K. (2021) ‘Accelerated algorithms for smooth convex-concave minimax problems with (O(1/k^2)) rate on squared gradient norm’. Proceedings of the 38th International Conference on Machine Learning.

[2] Abe, K., Sakamoto, M., Ariu, K., and Iwasaki, A. Boosting perturbed gradient ascent for last-iterate convergence in games. arXiv preprint arXiv:2410.02388, 2024